# FAN: FOURIER ANALYSIS NETWORKS

## ABSTRACT

Despite the remarkable success achieved by neural networks, particularly those represented by MLP and Transformer, we reveal that they exhibit potential flaws in the modeling and reasoning of periodicity, i.e., they exhibit satisfactory performance within the domain of training period, but struggle to generalize to out of the domain (OOD). The inherent cause lies in the way that they tend to memorize the periodic data rather than genuinely understand the underlying principles of periodicity. In fact, periodicity is essential to various forms of reasoning and generalization, underpinning predictability across natural and engineered systems through recurring patterns in observations. In this paper, we propose FAN, a novel network architecture based on Fourier Analysis, which empowers the ability to efficiently model and reason about periodic phenomena, meanwhile maintaining general-purpose ability. By introducing Fourier Series, periodicity is naturally integrated into the structure and computational processes of FAN. On this basis, FAN is defined following two core principles: 1) its periodicity modeling capability scales with network depth and 2) the periodicity modeling available throughout the network, thus achieving more effective expression and prediction of periodic patterns. FAN can seamlessly replace MLP in various model architectures with fewer parameters and FLOPs, becoming a promising substitute to traditional MLP. Through extensive experiments, we demonstrate the superiority of FAN in periodicity modeling tasks, and the effectiveness and generalizability of FAN across a range of real-world tasks, including symbolic formula representation, time series forecasting, language modeling, and image recognition.

## 1 INTRODUCTION

The flourishing of modern machine learning and artificial intelligence is inextricably linked to the revolutionary advancements in the foundational architecture of neural networks. For instance, multi-layer perceptron (MLP) (Rosenblatt, 1958; Haykin, 1998) plays a pivotal role in laying the groundwork for current deep learning models, with its expressive power guaranteed by the universal approximation theorem (Hornik et al., 1989). Recent claims about the impressive performance of large models on various tasks are typically supported by Transformer architecture (Vaswani et al., 2017; Touvron et al., 2023; OpenAI, 2023). In this context, the community's enthusiasm for research on neural networks has never diminished. Some emerged neural networks demonstrate notable capabilities in specific fields (Gu & Dao, 2023; Liu et al., 2024), sparking widespread discussion within the community.

Beneath the surface of apparent prosperity, we uncover a critical issue that remains in existing neural networks: *they struggle to model the periodicity from data, especially in OOD scenarios.* We showcase this issue through an empirical study as illustrated in Figure 1. The results indicate that existing neural networks, including MLP (Rosenblatt, 1958), KAN (Liu et al., 2024), and Transformer (Vaswani et al., 2017), face difficulties in fitting periodic functions, even on a simple sine function. Although they demonstrate proficiency in interpolation within the domain of training data, they tend to falter when faced with extrapolation challenges of test data, especially in OOD scenarios. Therefore, their generalization capacity is primarily dictated by the scale and diversity of the training data, rather than by the learned principles of periodicity to perform reasoning. We argue that periodicity is an essential characteristic in various forms of reasoning and generalization, as it provides a basis for predictability in many natural and engineered systems by leveraging recurring patterns in observations. In fact, real-world tasks inherently contain many periodic and non-periodic

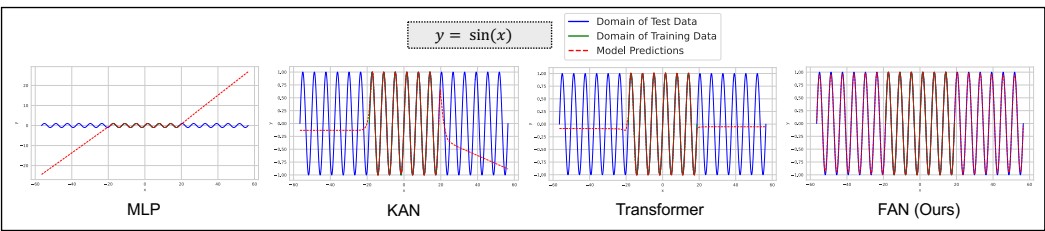

Figure 1: The performance of different neural networks within and outside the domain of their training data for the sine function, where $x$ is a scalar variable.

features, although some of them are hidden. The limitations of existing neural networks in capturing periodicity may impact their generalization performance, especially in OOD scenarios.

In this paper, we investigate a key research problem: *How to enable neural networks to model periodicity?* One core reason existing neural networks fail to model periodicity is that they heavily rely on data-driven optimization without explicit mechanisms to understand the underlying principles in the data. To this end, we propose a Fourier Analysis Network (FAN), a novel neural network framework based on Fourier Analysis. By leveraging the power of Fourier Series, we enable the neural network to capture and encode periodic patterns, offering a way to model the general principles from the data. Moreover, FAN is built upon two core principles: the first ensures that its periodic modeling capacity scales with network depth, while the second guarantees periodic modeling available throughout the network. FAN not only exhibits exceptional capabilities in periodicity modeling but also demonstrates competitive or superior effects on general tasks, which holds great potential as a substitute to traditional MLP.

To verify the effectiveness of FAN, we conduct extensive experiments from two main aspects: periodicity modeling and application of real-world tasks. 1) For periodicity modeling, FAN achieves significant improvements in fitting both basic and complex periodic functions, compared to existing neural networks (including MLP, KAN, and Transformer), particularly in OOD scenarios. 2) FAN demonstrates superior performance in real-world tasks, including symbolic formula representation, time series forecasting, language modeling, and image recognition. The experimental results indicate that FAN outperform baselines (including MLP, KAN, and Transformer) for symbolic formula representation task, and Transformer with FAN surpasses the competing models (including Transformer, LSTM, and Mamba), for time series forecasting and language modeling tasks. Moreover, FAN also shows effectiveness on standard CNN, especially in OOD scenarios, for image recognition tasks. As a promising substitute to MLP, FAN improves the model's generalization performance meanwhile reducing the number of parameters and floating point of operations (FLOPs) employed. We believe FAN is promising to be an important part of the fundamental model backbone.

## 2 PRELIMINARY KNOWLEDGE

Fourier Analysis (Stein & Weiss, 1971; Duoandikoetxea, 2024) is a mathematical framework that decomposes functions into their constituent frequencies, revealing the underlying periodic structures within complex functions. At the heart of this analysis lies Fourier Series (Tolstov, 2012), which expresses a periodic function as an infinite sum of sine and cosine terms. Mathematically, for a function $f(x)$, its Fourier Series expansion can be represented as:

$$f(x) = a_0 + \sum_{n=1}^{\infty} \left( a_n \cos\left(\frac{2\pi nx}{T}\right) + b_n \sin\left(\frac{2\pi nx}{T}\right) \right), \tag{1}$$

where $T$ is the period of the function, and the coefficients $a_n$ and $b_n$ are determined by integrating the function over one period:

$$a_n = \frac{1}{T} \int_0^T f(x) \cos\left(\frac{2\pi nx}{T}\right) dx, \quad b_n = \frac{1}{T} \int_0^T f(x) \sin\left(\frac{2\pi nx}{T}\right) dx. \tag{2}$$

The power of Fourier Series lies in its ability to represent a wide variety of functions, including non-periodic functions through periodic extensions, enabling the extraction of frequency components. Building on this mathematical foundation, FAN aims to embed the periodic characteristics directly into network architecture, enhancing generalization capabilities and performance on various tasks, particularly in scenarios requiring the identification of patterns and regularities.

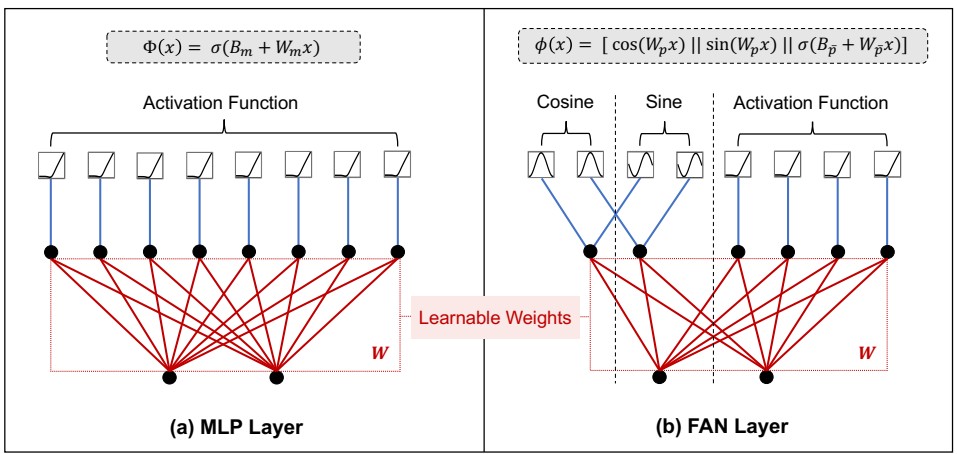

Figure 2: Illustrations of MLP layer $\Phi(x)$ vs. FAN layer $\phi(x)$.

## 3 FOURIER ANALYSIS NETWORK (FAN)

In this section, we first construct a simple neural network modeled by the formula of Fourier Series, and then on this basis, we design FAN and provide its details. Finally, we discuss the difference between the FAN layer and the MLP layer.

Consider a task involving input-output pairs $\{x_i, y_i\}$, with the objective of identifying a function $f(x) : \mathbb{R}^{d_x} \rightarrow \mathbb{R}^{d_y}$ that approximates the relationship such that $y_i \approx f(x_i)$ for all $x_i$, where $d_x$ and $d_y$ denote the dimensions of $x$ and $y$, respectively. To build a simple neural network $f_S(x)$ that represents Fourier Series expansion of the function, specifically $\mathcal{F}\{f(x)\}$, as described in Eq. (1), we can express $f_S(x)$ as follows:

$$
\begin{aligned}
f_S(x) &\triangleq a_0 + \sum_{n=1}^{N} \left( a_n \cos\left(\frac{2\pi n x}{T}\right) + b_n \sin\left(\frac{2\pi n x}{T}\right) \right), \\
&\overset{(\text{I})}{=} a_0 + \sum_{n=1}^{N} \left( w_n^c \cos\left(w_n^{\text{in}} x\right) + w_n^s \sin\left(w_n^{\text{in}} x\right) \right), \\
&\overset{(\text{II})}{=} B + [w_1^c, w_2^c, \cdots, w_n^c] \cos([w_1^{\text{in}}||w_2^{\text{in}}||\cdots||w_n^{\text{in}}]x) \\
&\quad + [w_1^s, w_2^s, \cdots, w_n^s] \sin([w_1^{\text{in}}||w_2^{\text{in}}||\cdots||w_n^{\text{in}}]x) \\
&= B + W_c \cos(W_{\text{in}}x) + W_s \sin(W_{\text{in}}x), \\
&\overset{(\text{III})}{=} B + W_{\text{out}}[\cos(W_{\text{in}}x)||\sin(W_{\text{in}}x)],
\end{aligned}
\tag{3}
$$

where $B \in \mathbb{R}^{d_y}, W_{\text{in}} \in \mathbb{R}^{N \times d_x}$, and $W_{\text{out}} \in \mathbb{R}^{d_y \times 2N}$ are learnable parameters, (I) follows that the computation of $a_n$ and $b_n$ computed via Eq. (2) is definite integral, (II) and (III) follows the equivalence of the matrix operations, $[\cdot||\cdot]$ and $[\cdot, \cdot]$ denotes the concatenation along the first and second dimension, respectively.

To fully leverage the advantages of deep learning, we can stack the aforementioned network $f_S(x)$ to form a deep network $f_D(x)$, where the $i$-th layer, denoted as $l_i(x)$, retains the same structural design as $f_S(x)$. Therefore, $f_D(x)$ can be formulated as:

$$
f_D(x) = l_L \circ l_{L-1} \circ \cdots \circ l_1 \circ x,
\tag{4}
$$

where $l_1 \circ x$ denotes the application of the left function $l_1$ to the right input $x$, that is $l_1(x)$. However, we discover that the direct stacking of $f_S(x)$ results in the primary parameters of the model $f_D(x)$ focusing on learning the angular frequency ($\omega_n = \frac{2\pi n}{T}$), thereby neglecting the learning of the Fourier coefficients ($a_n$ and $b_n$), as follows:

$$
\begin{aligned}
f_D(x) &= l_L(l_{L-1} \circ l_{L-2} \circ \cdots \circ l_1 \circ x) \\
&= B^L + W_{\text{out}}^L [\cos(W_{\text{in}}^L(l_{1:L-1} \circ x)) || \sin(W_{\text{in}}^L(l_{1:L-1} \circ x))]
\end{aligned}
\tag{5}
$$

where $l_{1:L-1} \circ x$ is defined as $l_{L-1} \circ l_{L-2} \circ \cdots \circ l_1 \circ x$, $W_{\text{in}}^L(l_{1:L-1} \circ x)$ is used to approximate the angular frequencies, and $W_{\text{out}}^L$ is used to approximate the Fourier coefficients. Therefore, the capacity of $f_D(x)$ to fit the Fourier coefficients is independent of the depth of $f_D(x)$, which is an undesirable outcome.

To this end, we design FAN based on the following principles: 1) the capacity of FAN to represent the Fourier coefficients should be positively correlated to its depth; 2) the output of any hidden layer can be employed to model periodicity using Fourier Series through the subsequent layers. The first one enhances the expressive power of FAN for periodicity modeling by leveraging its depth, while the second one ensures that the features of FAN's intermediate layers are available to perform periodicity modeling.

Suppose we decouple $f_S(x)$ as follows:

$$
f_S(x) = f_{out} \circ f_{in} \circ x,
\tag{6}
$$

where

$$
f_{in}(x) = [\cos(W_{\text{in}}x) || \sin(W_{\text{in}}x)],
\tag{7}
$$
$$
f_{out}(x) = B + W_{\text{out}}x.
\tag{8}
$$

To satisfy both principles, the inputs of the intermediate layers in FAN necessitate to employ $f_{in}$ and $f_{out}$ simultaneously, rather than applying them sequentially.

Finally, FAN is designed on this basis, with the FAN layer $\phi(x)$ defined as below:

$$
\phi(x) \triangleq [\cos(W_p x) || \sin(W_p x) || \sigma(B_{\bar{p}} + W_{\bar{p}} x)],
\tag{9}
$$

where $W_p \in \mathbb{R}^{d_x \times d_p}, W_{\bar{p}} \in \mathbb{R}^{d_x \times d_{\bar{p}}}$, and $B_{\bar{p}} \in \mathbb{R}^{d_{\bar{p}}}$ are learnable parameters (with the hyperparameters $d_p$ and $d_{\bar{p}}$ indicating the first dimension of $W_p$ and $W_{\bar{p}}$, respectively), the layer output $\phi(x) \in \mathbb{R}^{2d_p + d_{\bar{p}}}$, and $\sigma$ denotes the activation function, which can further enhance its expressive power for periodicity modeling.

The entire FAN is defined as the stacking of the FAN layer $\phi(x)$:

$$
\text{FAN}(x) = \phi_L \circ \phi_{L-1} \circ \cdots \circ \phi_1 \circ x,
\tag{10}
$$

where

$$
\phi_l(x) = \begin{cases} [\cos(W_p^l x) || \sin(W_p^l x) || \sigma(B_{\bar{p}}^l + W_{\bar{p}}^l x)], & \text{if } l < L, \\ B^L + W^L x, & \text{if } l = L, \end{cases}
\tag{11}
$$

Table 1: Comparison of MLP layer and FAN layer, where $d_p$ is a hyperparameter of FAN layer and defaults to $\frac{1}{4}d_{\text{output}}$ in this paper, $d_{\text{input}}$ and $d_{\text{output}}$ denote the input and output dimensions of the neural network layer, respectively. In our evaluation, the FLOPs for any arithmetic operations are considered as 1, and for Boolean operations as 0.

| | MLP Layer | FAN layer |
|---|---|---|
| Formula | $\Phi(x) = \sigma(B_m + W_m x)$ | $\phi(x) = [\cos(W_p x) || \sin(W_p x) || \sigma(B_{\bar{p}} + W_{\bar{p}} x)]$ |
| Num of Params | $(d_{\text{input}} \times d_{\text{output}}) + d_{\text{output}}$ | $\left(1 - \frac{d_p}{d_{\text{output}}}\right) \times ((d_{\text{input}} \times d_{\text{output}}) + d_{\text{output}})$ |
| FLOPs | $2 \times (d_{\text{input}} \times d_{\text{output}})$ $+\text{FLOPs}_{\text{non-linear}} \times d_{\text{output}}$ | $\left(1 - \frac{d_p}{d_{\text{output}}}\right) \times 2 \times (d_{\text{input}} \times d_{\text{output}})$ $+\text{FLOPs}_{\text{non-linear}} \times d_{\text{output}}$ |

The illustrations of the MLP layer $\Phi(x)$ vs. the FAN layer $\phi(x)$ are shown in Figure 2. Note that the FAN layer $\phi(x)$ computed via Eq. (9) can seamlessly replace the MLP layer $\Phi(x)$ computed via Eq. (12) in various models with fewer parameters and FLOPs, achieved by sharing the parameters and computation of Sin and Cos parts. The number of parameters and FLOPs of the FAN layer compared to the MLP layer are presented in Table 1.

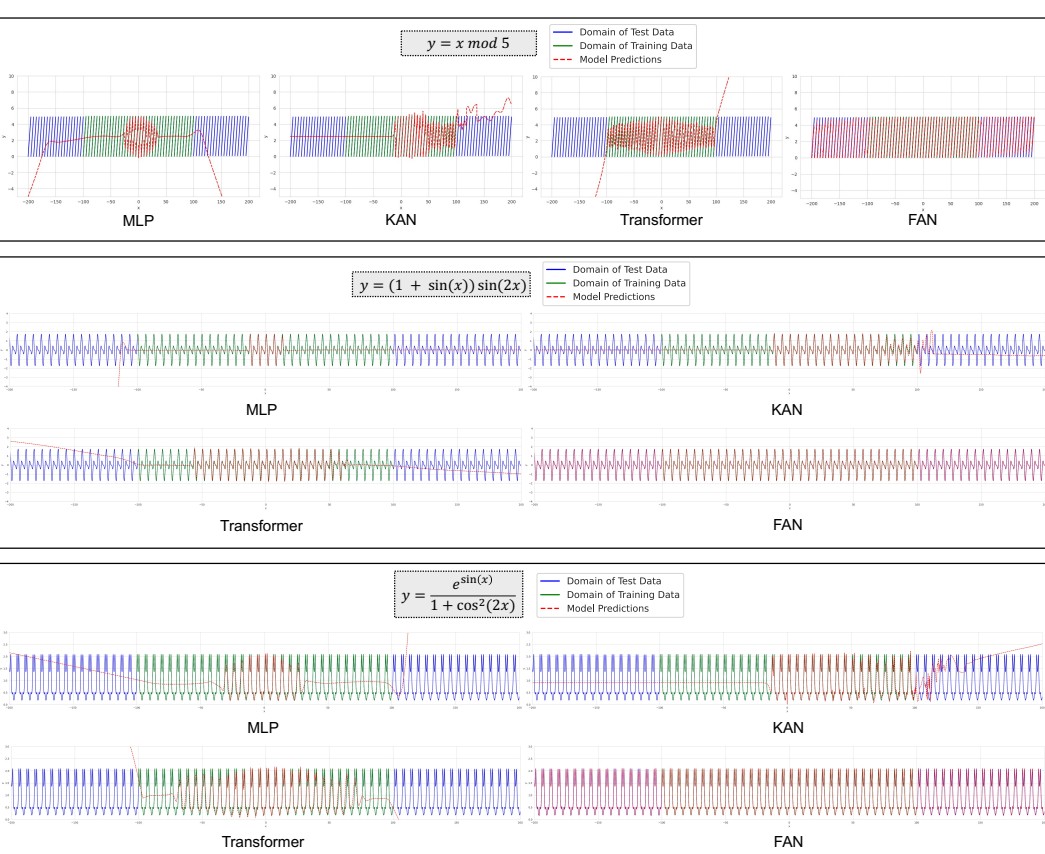

Figure 3: The performance of FAN in periodicity modeling compared to MLP, KAN, and Transformer, where the green line represents the test data within the domain of the training data, while the blue line represents the test data outside the domain of the training data.

## 4 EXPERIMENTS

In this section, we first introduce the baselines and implementation details of our experiments. Second, we verify the superiority of FAN in periodicity modeling tasks (Section 4.1). Third, we demonstrate the effectiveness and generalizability of FAN across a range of real-world tasks, including symbolic formula representation (Section 4.2), time series forecasting (Section 4.3), language modeling (Section 4.4), and image recognition (Section 4.5). Finally, we conduct further analysis on FAN's running time and hyperparameter impact. (Section 4.6).

**Baselines.** In our experiments, we mainly compare FAN with the following baselines: 1) **MLP** (Rosenblatt, 1958), 2) **Transformer** (Vaswani et al., 2017), 3) **KAN** (Liu et al., 2024), 4) **LSTM** (Hochreiter & Schmidhuber, 1997), 5) **Mamba** (Gu & Dao, 2023), 6) **CNN** (LeCun et al., 1998). Details of the baselines are given in Appendix F. Moreover, we also include the following variants of FAN into our comparisons: I) **FAN (Gated)**: a variant of FAN that adds gates to control the tendency of the layer, with the formula defined as $\phi_g(x) = [g \cdot \cos(W_p x) || g \cdot \sin(W_p x) || (1-g) \cdot \sigma(B_{\bar{p}} + W_{\bar{p}} x)]$, where $g$ is a learnable parameter. II) **Transformer with FAN and Transformer with FAN (Gated)**: we replace each MLP layer in Transformer with the FAN layer computed via Eq. (9) and the layer

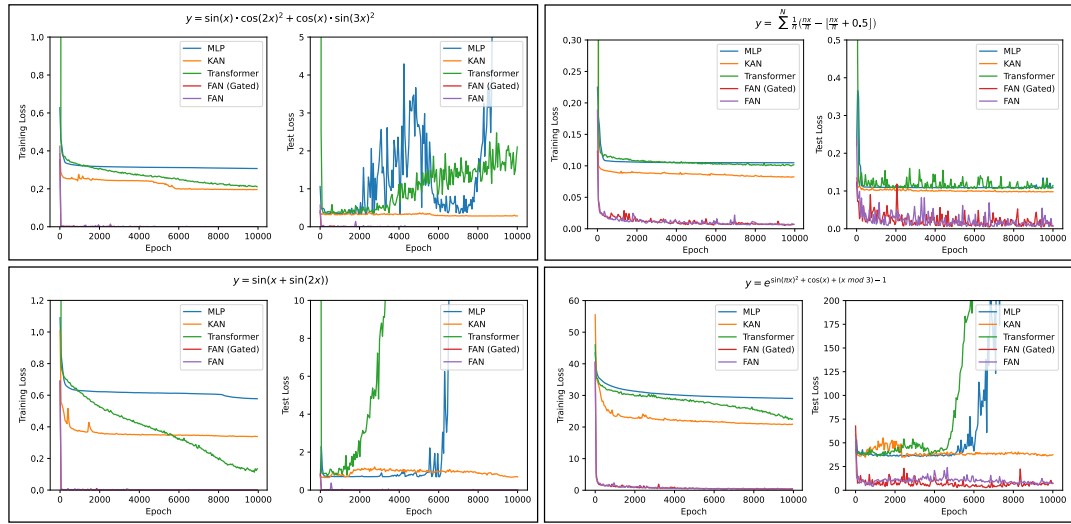

Figure 4: Comparison of training and test losses for different models on the tasks of learning complex periodic functions.

of FAN (Gated), respectively. III) **CNN with FAN**: similarly, we replace each MLP layer in CNN with the FAN layer.

**Implementation Details.** We conduct our experiments on a single GPU of Tesla A100-PCIe-40G. Unless otherwise specified, we use the following hyperparameters in the experiments. The model architecture consists of 3-12 layers, the activation function $\sigma$ is set to GELU (Hendrycks & Gimpel, 2016), and the dimension of the projection matrix $W_p$ is set to $d_p = \frac{1}{4}d_h$, where $d_h$ denotes the dimension of the hidden layers. We employ the AdamW optimizer (Loshchilov & Hutter, 2019) for the model's training process. More experimental details and comprehensive setups of each task can be found in Appendix C.

### 4.1 PERIODICITY MODELING

**Setup.** In periodic modeling tasks, we select periodic functions with practical significance and compare the models' performance in learning the underlying principles of periodicity. Specifically, we generate data from periodic functions over a large domain, using a portion of this domain as training data and the entire domain as test data, i.e., a part of test data would be out of the domain of training data. In this task, we compare FAN and its variant, FAN (Gated), with MLP, KAN, and Transformer. The input of each task is a scalar.

**Results.** Figure 3 illustrates the performance of FAN and other baselines in periodicity modeling. The results indicate that existing neural networks, including MLP, KAN, and Transformers, exhibit notable deficiencies in their ability to model periodicity. Although they attempt to fit these periodic functions, their ability limits their performance in modeling a large domain of periodicity, including the test data within and outside the domain of the training data. In contrast, FAN significantly outperforms the baselines in all these tasks of periodicity modeling. Moreover, FAN performs exceptionally well on the test data both within and outside the domain, indicating that it is genuinely modeling periodicity rather than merely memorizing the training data.

We also analyze the training process of different models on the tasks of learning complex periodic functions, as illustrated in Figure 4, which leads to the following findings. 1) FAN far exceeds the other baselines in both convergence speed and final effects. 2) In comparison to FAN, FAN (Gated) often achieves faster convergence, but the final performance remains comparable. 3) Although the baselines show stabilization or gradual reductions in training loss as the number of epochs increases, their modeling may have diverged considerably from the distribution of the test data, resulting in a

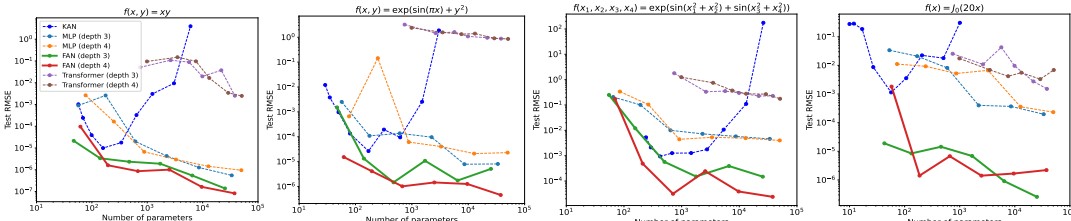

Figure 5: Comparisons of FAN with the baselines, including MLP, KAN, and Transformer, across varying numbers of parameters on symbolic formula representation tasks.

sharp increase in test loss. This phenomenon further demonstrates the shortcomings of these models in capturing periodicity.

### 4.2 SYMBOLIC FORMULA REPRESENTATION

**Setup.** Symbolic formula representation is a common task in both mathematics and physics. We follow the experiments conducted in KAN's paper (Liu et al., 2024), adhering to the same tasks, data, hyperparameters, and baselines. In addition to the original baselines, we also include Transformer for comparison in this task.

**Results.** Figure 5 demonstrates the performance of different models applied to four common functions in mathematics and physics. From Figure 5, we can observe that while KAN remains competitive with FAN when the number of parameters is small, its performance declines clearly as the number of parameters increases, which exhibits a U-shaped trend (Liu et al., 2024). In contrast, as the number of parameters becomes large, FAN consistently outperforms the other baselines, including MLP, KAN, and Transformer, in fitting these functions, despite many of these functions being only partially periodic or entirely non-periodic. This may be attributed to FAN's ability to capture and model both periodic and non-periodic features and the advantages of fewer parameters. These results indicate that although FAN enhances its ability to model periodicity, it does not compromise its capacity to fit non-periodic functions.

Table 2: Performance of different sequence models on time series forecasting tasks, where Input Length = 96, the **bold** values indicate the lowest value on each row, and the improve means the relative improvements of using FAN and FAN (Gated) based on Transformer.

| Dataset | Output Length | LSTM (12.51 M) | | Mamba (12.69 M) | | Transformer (12.12 M) | | Transformer with FAN (11.06 M) | | | |
|---|---|---|---|---|---|---|---|---|---|---|---|
| | | | | | | | | Gated | | Default | |
| | | MSE ↓ | MAE ↓ | MSE ↓ | MAE ↓ | MSE ↓ | MAE ↓ | MSE ↓ | MAE ↓ | MSE ↓ | MAE ↓ |
| Weather | 96 | 1.069 | 0.742 | 0.552 | 0.519 | 0.413 | 0.438 | **0.292** | **0.380** | 0.313 | 0.431 |
| | 192 | 1.090 | 0.778 | 0.700 | 0.595 | 0.582 | 0.540 | 0.535 | 0.550 | **0.472** | **0.525** |
| | 336 | 0.992 | 0.727 | 0.841 | 0.667 | 0.751 | 0.626 | **0.637** | 0.602 | 0.719 | **0.581** |
| | 720 | 1.391 | 0.892 | 1.171 | 0.803 | 0.967 | 0.715 | 0.845 | 0.706 | **0.732** | **0.670** |
| Exchange | 96 | 0.938 | 0.794 | 0.908 | 0.748 | 0.777 | 0.681 | 0.685 | 0.644 | **0.657** | **0.623** |
| | 192 | 1.241 | 0.899 | 1.328 | 0.925 | 1.099 | 0.800 | 0.998 | 0.757 | **0.968** | **0.741** |
| | 336 | 1.645 | 1.048 | 1.512 | 0.992 | 1.614 | 1.029 | 1.511 | 0.961 | **1.266** | **0.905** |
| | 720 | 1.949 | 1.170 | 2.350 | 1.271 | 2.163 | 1.204 | **1.658** | **1.104** | 2.063 | 1.205 |
| Traffic | 96 | 0.659 | 0.359 | 0.666 | 0.377 | 0.656 | 0.357 | 0.647 | 0.355 | **0.643** | **0.347** |
| | 192 | 0.668 | 0.360 | 0.671 | 0.381 | 0.672 | 0.363 | **0.649** | **0.353** | 0.657 | 0.354 |
| | 336 | **0.644** | **0.342** | 0.665 | 0.374 | 0.673 | 0.360 | 0.665 | 0.358 | 0.656 | 0.353 |
| | 720 | **0.654** | **0.351** | 0.662 | 0.364 | 0.701 | 0.380 | 0.682 | 0.369 | 0.673 | 0.363 |
| ETTh | 96 | 0.999 | 0.738 | 0.860 | 0.697 | 1.139 | 0.853 | **0.842** | 0.736 | 0.873 | **0.707** |
| | 192 | 1.059 | 0.759 | **0.849** | **0.700** | 1.373 | 0.932 | 0.885 | 0.748 | 0.914 | 0.741 |
| | 336 | 1.147 | 0.820 | 1.005 | 0.745 | 1.261 | 0.924 | **0.980** | **0.770** | 0.999 | 0.793 |
| | 720 | 1.206 | 0.847 | **0.994** | **0.758** | 1.056 | 0.819 | 1.002 | 0.798 | 1.031 | 0.818 |
| Average (Improve) | – | 1.083 | 0.726 | 1.002 | 0.668 | 0.994 | 0.689 | **0.845** ↓15.0% | 0.637 ↓7.6% | 0.852 ↓14.3% | **0.635** ↓7.9% |

## 4.3 TIME SERIES FORECASTING

**Setup.** Time series forecasting plays a critical role in various real-world applications. In our experiments, we employ four public datasets of this task to assess the model performance on time series forecasting, including Weather (Wu et al., 2021a), Exchange (Lai et al., 2018), Traffic (Wu et al., 2021a), and ETTh (Zhou et al., 2021) datasets. For each dataset, we input 96 previous time steps and forecast the subsequent time steps of {96, 192, 336, 720}. In this task, we choose the sequence models as baselines, including LSTM, Mamba, Transformer, Transformer with FAN , and Transformer with FAN (Gated).

**Results.** As presented in Table 2, we compare the performance of Transformer with FAN and other sequence models for time series forecasting tasks on four public datasets. In most cases, Transformer with FAN and its gated version achieves the best performance on these tasks, compared to LSTM, Mamba, and the standard Transformer. The improvements of Transformer with FAN and FAN (Gated) over the standard Transformer are notable, with the average relative improvements ranging from 14.3% to 15.0% for MSE and from 7.6% to 7.9% for MAE. These results suggest that incorporating explicit periodic pattern encoding within neural networks improves time series forecasting performance in real-world applications.

## 4.4 LANGUAGE MODELING

**Setup.** Language modeling is a fundamental task in natural language processing. In this experiment, we conduct language modeling using the SST-2 (Socher et al., 2013) dataset and evaluate the model's performance on its test set, as well as on the related datasets such as IMDB (Maas et al., 2011), Sentiment140 (Sahni et al., 2017), and Amazon Reviews (Linden et al., 2003). These four classic datasets all belong to the field of sentiment analysis. In this task, the comparisons are between Transformer with FAN and FAN (Gated), along with other sequence models, including LSTM, Mamba, and Transformer.

Table 3: Performance of different sequence models on language modeling tasks, where the models are trained on the training set of SST-2 and evaluated on the other datasets, the **bold** value indicates the best performance on each column, the ***bold italic*** indicates the best performance other than Transformer with FAN and FAN (Gated), and the improvements represent our relative improvements of using FAN  based on Transformer.

| Model | Num of Params | SST-2 (test) | | IMDB | | Sentiment140 | | Amazon Reviews | |
|---|---|---|---|---|---|---|---|---|---|
| | | Loss ↓ | Acc ↑ | Loss ↓ | Acc ↑ | Loss ↓ | Acc ↑ | Loss ↓ | Acc ↑ |
| LSTM | 120.14M | 0.4760 | 0.8060 | 0.6449 | 0.6438 | 0.8026 | ***0.5979*** | 0.5791 | 0.7152 |
| Mamba | 129.73M | 0.4335 | 0.7959 | 0.6863 | 0.6203 | **0.7871** | 0.5874 | 0.6163 | 0.6719 |
| Transformer | 109.48M | ***0.4297*** | ***0.8119*** | ***0.5649*** | ***0.6994*** | 0.8891 | 0.5779 | ***0.5563*** | ***0.7155*** |
| w/ FAN (Gated) | 95.33M | 0.4250 | 0.8039 | 0.5817 | 0.7012 | 0.7941 | **0.6194** | 0.4835 | 0.7689 |
| w/ FAN | 95.32M | **0.4094** | **0.8154** | **0.5225** | **0.7398** | 0.8257 | 0.6093 | **0.4748** | **0.7763** |
| Improvements | ↓ 14.16M | ↓ 4.72% | ↑ 0.43% | ↓ 7.51% | ↑ 5.78% | ↓ 7.13% | ↑ 5.43% | ↓ 14.65% | ↑ 8.50% |

**Results.** We report the performance comparison between different sequence models across four public sentiment analysis datasets, as shown in Table 3. The results indicate that Transformer with FAN achieves clear improvements compared to the standard Transformer and other baselines, such as LSTM and Mamba, especially for zero-shot OOD performance on IMDB, Sentiment140, and Amazon Reviewers datasets. Using FAN achieves the relative improvements up to 14.65% and 8.50% in terms of Loss and Accuracy respectively, while reducing the number of parameters by about 14.16M. The result indicates the potential of periodicity modeling to enhance both effectiveness and generalization on cross-domain language modeling and sentiment analysis tasks.

## 4.5 IMAGE RECOGNITION

**Setup.** Image recognition is a key computer vision task where image content is identified and categorized. Our evaluation contains four public benchmarks of image recognition: MNIST (LeCun

et al., 2010), MNIST-M (Ganin et al., 2016), Fashion-MNIST (Xiao et al., 2017), and Fashion-MNIST-C (Weiss & Tonella, 2022), where MNIST is used for digit recognition, Fashion-MNIST assesses clothing classification, MNIST-M and Fashion-MNIST-C are their variant for robustness.

**Results.** We also apply FAN to image recognition tasks on four classic benchmarks, as shown in Table 4. Experimental results show that using FAN outperforms the standard CNN in most cases for the optimal Accuracy, Accuracy, and OOD Accuracy, as well as achieves clear improvements in the optimal OOD Accuracy. We believe that there are some latent periodic features in image recognition tasks, and FAN's ability to model these periodic features can help CNN achieve competitive or superior performance, especially in OOD scenarios.

Table 4: Results on image recognition tasks. Accuracy* means best Accuracy, Accuracy means Accuracy at the last epoch, and OOD Accuracy means Accuracy on other paired datasets. **Bold** values indicate the highest value between CNN and CNN w/ FAN under the same setting.

| Dataset | Accuracy* ↑ | | OOD Accuracy* ↑ | | Accuracy ↑ | | OOD Accuracy ↑ | |
|---|---|---|---|---|---|---|---|---|
| | CNN | w/ FAN | CNN | w/ FAN | CNN | w/ FAN | CNN | w/ FAN |
| MNIST | 99.63 | **99.67** | 28.85 | **30.3** | 99.55 | **99.64** | **22.12** | 21.64 |
| MNIST-M | **94.52** | 94.23 | 82.85 | **83.55** | **94.29** | 94.22 | 80.07 | **81.44** |
| Fashion-MNIST | 94.15 | **94.47** | 49.82 | **51.88** | 94.05 | **94.21** | 48.08 | **50.3** |
| Fashion-MNIST-C | 88.61 | **88.82** | 91.45 | **91.59** | **88.6** | 88.59 | 91.41 | **91.47** |

## 4.6 FURTHER ANALYSIS OF FAN

**Runtime of FAN.** We analyze the actual running time of the FAN Layer compared to the MLP Layer with different input and output dimensions, as shown in Table 5. The experimental re-

Table 5: Comparison of actual runtime between FAN and MLP.

| | 1024×1024 | 2048×2048 | 4096×4096 | 8192×8192 |
|---|---|---|---|---|
| MLP | **0.064 ms** | **0.114 ms** | 0.212 ms | 0.938 ms |
| FAN | 0.128 ms | 0.133 ms | **0.211 ms** | **0.704 ms** |

sults show that MLPs exhibit smaller runtimes when the input and output sizes are small, due to PyTorch's optimization of MLP. However, as the input and output sizes continue to increase, matrix computations become the main contributor to runtime. At this point, FAN's fewer parameters and reduced FLOPs begin to show significant advantages. Note that FAN can be further optimized from the underlying implementation.

**The impact of hyperparameter $d_p$.** In our experiments, we fix the hyperparameter $d_p = \frac{1}{4} d_h$ intuitively for FAN, where $d_h$ denotes the dimension of the hidden layers. As shown in Figure 7 of Appendix, we investigate the impact of varying $d_p$ empirically on task performance by changing itself. The results indicate that performance initially improves as $d_p$ increases, but then decreases beyond a certain point. This trend may be attributed to the number of potential periodic features specific to each task. Furthermore, there remains room for further improvements with the better hyperparameter setup of $d_p$.

## 5 RELATED WORK

In this section, we outline the two most relevant directions and associated papers of this work.

**Learning Periodicity with Neural Networks.** Periodic functions are one of the most basic functions of importance to human society and natural science (Newton, 1687; Osborn & Sensier, 2002; Kwasnicki, 2008; De Groot & Franses, 2012; Zhang et al., 2017). However, commonly used neural networks, such as MLPs and transformers, struggle with modeling periodicity. This limitation is attributed to the lack of inherent "periodicity" in their inductive biases. Some previous works (Silvescu, 1999; Liu, 2013; Parascandolo et al., 2016; Uteuliyeva et al., 2020) proposed merely using standard periodic functions themselves or their linear combinations as activation functions, which

only work well on some shallow and simple models. On this basis, work (Liu et al., 2020) introduced the Snake function, i.e., $x + \sin^2(x)$, as the activation function. However, we observed that it can fit periodic functions to a certain extent, but its effect is limited especially for OOD scenarios, as demonstrated in Appendix D. Therefore, although some previous studies have attempted to integrate periodic information into neural networks, their actual performance and range of applications remain heavily constrained.

**Fourier-based Neural Network.** Previous studies have explored Fourier-based neural networks to enhance the computational tasks (Zuo & Cai, 2005; Tan, 2006; Zuo et al., 2008; Li et al., 2021b; Chen et al., 2022). Fourier Neural Networks (Silvescu, 1999; Ngom & Marin, 2021) are shallow feedforward networks that employ cosine activation functions to map inputs to their Fourier decompositions. Work (Lee et al., 2021) directly utilized the Fourier Series constructed by a shallow neural network for generating periodic signals. In addition, work (Jiang et al., 2022) introduces Fourier Series at the end of models to embed periodic components within the network. These approaches generally possess a similar principle as Eq. (3), using a neural network to simulate the formula of Fourier Series. However, this leads to the same problem as in Eq. (5), i.e., they are hard to serve as building blocks for deep neural networks, which limits these approaches' capabilities.

In this paper, we design FAN to address these challenges, which performs exceptionally well on periodicity modeling and a range of real-world tasks.

## 6 DISCUSSION

In this section, we mainly discuss the expressive power and application scope of FAN as follows.

First, FAN theoretically possesses the same expressive power as MLP as it also adheres to the universal approximation theorem, which ensures its capacity for functional approximation (refer to Appendix E for the detailed explanation). Moreover, FAN introduces an important enhancement by explicitly incorporating periodicity, a feature absent in traditional MLPs. Through this design, FAN not only retains the capabilities of MLP but also enhances its ability to capture periodic characteristics in data. For periodic tasks and some non-periodic tasks that are partially periodic, FAN leverages its effective periodicity modeling ability to yield better results. Therefore, FAN can be seen as a promising alternative to MLP.

Second, beyond tasks that explicitly require periodicity modeling, FAN also has utility in a broader range of applications, which has been evidenced by our extensive experiments on real-world tasks, such as symbolic formula representation, time series forecasting, language modeling, and image recognition, where FAN achieve competitive or superior performance than MLP and other baselines. In fact, many machine learning tasks may harbor hidden forms of periodicity, even without explicitly including periodicity, such as mathematical operations and logic reasoning. If the neural network lacks the ability to model periodicity, it could impair its learning efficiency. From a deeper perspective, periodicity is not just a data feature but reflects a form of structural knowledge—one that allows for the transfer and reuse of abstract rules and principles across different contexts.

## 7 CONCLUSION

In this paper, we have proposed Fourier Analysis Network (FAN), a novel neural network architecture for tackling the problem of periodicity modeling, which utilizes Fourier Series to facilitate capturing the underlying principles within data and reasoning. Experimental results demonstrate that FAN can successfully fit a variety of both basic and complex periodic functions, whereas other approaches failed. Moreover, FAN and its combination with Transformer also exhibit superior performance in multiple real-world tasks, including symbolic formula representation, time series forecasting, language modeling, and image recognition tasks, outperforming existing neural networks such as MLP, KAN, Transformer, CNN, LSTM, and Mamba. These promising results, especially the stronger performance and the fewer parameters and FLOPs compared to MLP, suggest its potential to become a key component of foundational models. In the future, we aim to further increase the scale of FAN and expand its scope of application, reinforcing its role as a versatile and powerful building block in the machine learning landscape.

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

# A MLP

The MLP layer $\Phi(x)$ is defined as:

$$\Phi(x) = \sigma(B_m + W_m x), \tag{12}$$

where $B_m \in \mathbb{R}^{d_m}$ and $W_{\bar{p}} \in \mathbb{R}^{d_x \times d_m}$ are learnable parameters with the hyperparameter $d_m$ indicating the first dimension of $W_m$, $\sigma$ denotes the activation function, and MLP can be defined as the stacking of the MLP layer $\Phi(x)$:

$$\text{MLP}(x) = \Phi_L \circ \Phi_{L-1} \circ \cdots \circ \Phi_1 \circ x, \tag{13}$$

where

$$\Phi_l(x) = \left\{ \begin{array}{ll} \sigma(B_m^l + W_m^l x), & \text{if } l < L, \\ B^L + W^L x, & \text{if } l = L. \end{array} \right. \tag{14}$$

# B ADDITIONAL EXPERIMENTS

## B.1 ADDITIONAL EXPERIMENTS ON PERIODICITY MODELING TASKS.

More experimental results on periodicity modeling tasks are shown in Figure 6.

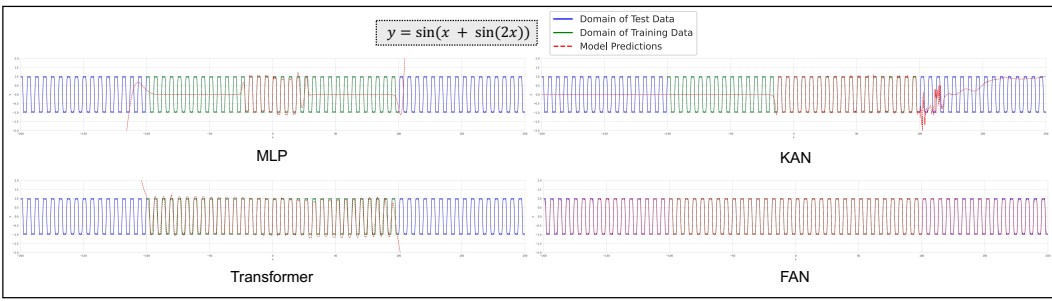

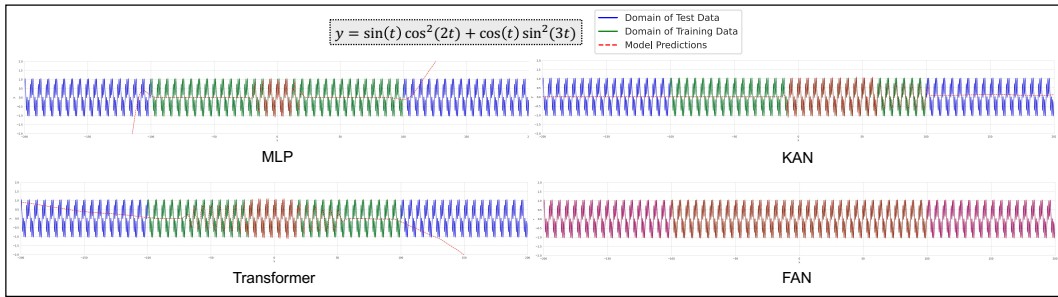

Figure 6: Additional Experiments on Periodicity Modeling Tasks.

## B.2 FAN FOR SOLVING SCIML PROBLEMS

We conduct experiments on the SciML problem that includes the Fourier function class following the work (Li et al., 2021a). The Burgers' equation, a non-linear partial differential equation, is frequently used in scientific computing to model shock waves and traffic flow, among other phenomena. The detailed error rate on Burgers' equation is listed in the Table 6. We can find that replacing the MLP Layer with FAN Layer in Fourier Neural Operator (FNO) (Li et al., 2021a) can achieve clear improvements on each setting of resolution $s$ of this task.

## B.3 COMPARISON WITH FREQUENCY-BASED MODELS IN TIME SERIES FORECASTING TASKS

To compare with frequency-based models in Time Series Forecasting tasks such as FEDformer (Zhou et al., 2022), we replace MLP with FAN in frequency-based models. We present the experi-

Table 6: The error rate on Burgers' equation. The values in the table represent the Average Relative Error for Burgers' equation with lower values indicating better performance.

| Model | $s = 256$ | $s = 512$ | $s = 1024$ | $s = 2048$ | $s = 4096$ | $s = 8192$ |
|---|---|---|---|---|---|---|
| FNO | 5.93% | 6.14% | 6.03% | 6.75% | 7.36% | 9.93% |
| FNO with FAN | **5.26%** | **5.17%** | **5.18%** | **6.73%** | **6.35%** | **7.06%** |

mental results in Table 7, where the results of FEDformer are cited from its paper directly. From the results, we can find that FEDformer with FAN can outperform FEDformer in almost all cases.

Table 7: Results of comparison with frequency-based models in time series forecasting tasks.

| Dataset | Len | FEDformer | | with FAN | |
|---|---|---|---|---|---|
| | | MSE | MAE | MSE | MAE |
| Traffic | 96 | 0.587 | 0.366 | **0.577** | **0.357** |
| | 192 | 0.604 | 0.373 | **0.601** | **0.366** |
| | 336 | 0.621 | 0.383 | **0.620** | **0.378** |
| | 720 | 0.626 | 0.382 | **0.619** | **0.370** |
| Exchange | 96 | 0.148 | 0.278 | **0.138** | **0.267** |
| | 192 | 0.271 | 0.380 | **0.261** | **0.371** |
| | 336 | **0.460** | **0.500** | 0.461 | 0.503 |
| | 720 | 1.195 | 0.841 | **1.159** | **0.827** |
| Electricity | 96 | 0.193 | 0.308 | **0.184** | **0.298** |
| | 192 | 0.201 | 0.315 | **0.199** | **0.313** |
| | 336 | 0.214 | 0.329 | **0.212** | **0.325** |
| | 720 | 0.246 | 0.355 | **0.239** | **0.347** |

### B.4 COMPARISON WITH DIRECTLY LEARNING THE COEFFICIENTS

We compare FAN with a baseline of directly learning the coefficients, which inputs $sin(x)$ and $cos(x)$ and then uses the MLP Layer instead of the FAN Layer to model the Fourier coefficients. In this setting, frequencies are fixed and only the coefficients are learned, which may limit the model's ability to capture patterns not aligned with these frequencies. Taking simple $f(x) = x \bmod 5$ as an example, this setting may not even converge at all, because the frequency of $x \bmod 5$ is inconsistent with $sin(x)$ and $cos(x)$. The experimental results of their loss are shown in Table 8.

Table 8: Comparison of FAN and directly learning the coefficients on fitting $f(x) = x \bmod 5$.

| Epoch | 50 | 100 | 150 | 200 |
|---|---|---|---|---|
| Directly learning the coefficients | 2.10 | 2.09 | 2.09 | 2.08 |
| FAN | 0.28 | 0.23 | 0.18 | 0.17 |

### B.5 EXPERIMENTS ON TIME SERIES FORECASTING WITH INSTANCE NORMALIZATION

We conduct experiments on time series forecasting tasks with instance normalization (Ulyanov et al., 2016), and the results are shown in Table 9. We find that applying instance normalization before the architecture can effectively improve the performance.

### B.6 THE INFLUENCE OF HYPERPARAMETERS $\mathbf{d_p}$

We evaluate the influence of hyperparameters $\mathbf{d_p}$ as shown in Figure 7.

Table 9: Results on time series forecasting tasks with instance normalization, where Input Length = 96, the **bold** values indicate the lowest value on each row, and the improve means the relative improvements of using FAN and FAN (Gated) based on Transformer.

| Dataset | Output Length | Transformer (12.12 M) | | Transformer with FAN (11.06 M) | | | |
| | | | | Gated | | Default | |
| | | MSE ↓ | MAE ↓ | MSE ↓ | MAE ↓ | MSE ↓ | MAE ↓ |
|---|---|---|---|---|---|---|---|
| Weather | 96 | 0.1772 | 0.2301 | 0.1864 | 0.2352 | **0.1756** | **0.2247** |
| | 192 | 0.2438 | 0.2844 | 0.2445 | 0.2834 | **0.2327** | **0.2760** |
| | 336 | **0.3077** | **0.3267** | 0.3156 | 0.3320 | 0.3118 | 0.3291 |
| | 720 | 0.4253 | 0.3982 | **0.3909** | **0.3782** | 0.4113 | 0.3906 |
| Exchange | 96 | 0.1433 | 0.2653 | **0.1157** | **0.2452** | 0.1436 | 0.2666 |
| | 192 | 0.2563 | **0.3552** | 0.2539 | 0.3611 | 0.2651 | 0.3757 |
| | 336 | 0.5273 | 0.5218 | **0.4329** | **0.4891** | 0.5092 | 0.5326 |
| | 720 | 1.7401 | 0.9273 | 1.5783 | 0.9303 | **1.0599** | **0.7657** |
| Traffic | 96 | 0.6160 | 0.3449 | **0.6030** | 0.3334 | 0.6109 | **0.3319** |
| | 192 | 0.6329 | 0.3479 | 0.6239 | 0.3404 | 0.6258 | **0.3370** |
| | 336 | 0.6369 | 0.3485 | 0.6416 | 0.3487 | **0.6200** | **0.3380** |
| | 720 | 0.6555 | 0.3577 | 0.6645 | 0.3574 | **0.6412** | **0.3525** |
| ETTh1 | 96 | **0.5339** | **0.4910** | 0.5503 | 0.5216 | 0.5378 | 0.4983 |
| | 192 | **0.5633** | **0.5209** | 0.5906 | 0.5346 | 0.5968 | 0.5265 |
| | 336 | 0.7576 | 0.5813 | **0.6640** | **0.5636** | 0.7525 | 0.5933 |
| | 720 | 0.7411 | 0.6177 | 0.7411 | **0.6066** | **0.7328** | 0.6142 |
| ETTh2 | 96 | 0.3881 | **0.4097** | 0.4082 | 0.4292 | **0.3833** | 0.4149 |
| | 192 | 0.5766 | 0.4999 | **0.4695** | **0.4514** | 0.5039 | 0.4640 |
| | 336 | 0.5782 | 0.5100 | 0.5556 | 0.5012 | **0.5417** | **0.4940** |
| | 720 | 0.5841 | 0.5230 | **0.5070** | **0.4943** | 0.5272 | 0.4951 |
| Average (Improve) | – | 0.554 | 0.444 | 0.526 ↓ 5.1% | 0.436 ↓ 1.9% | **0.509** ↓ 8.2% | **0.430** ↓ 3.2% |

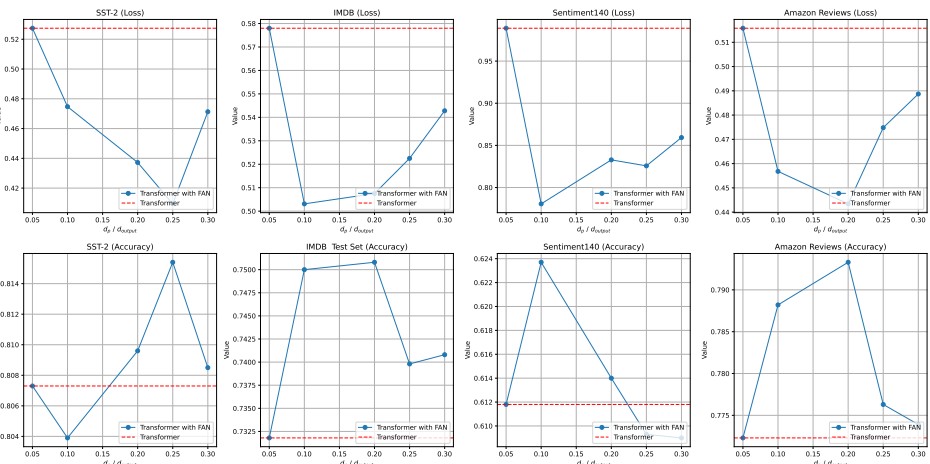

Figure 7: The influence of hyper-parameters $d_p$ on language modeling tasks. We use the red dashed line to represent the performance of the standard Transformer.

### B.7 THE EFFECTIVENESS OF THE FAN LAYER FOR DEEP NEURAL NETWORKS

We evaluate the effect of varying the number of FAN layers from 3 to 20 on periodicity modeling tasks, employing residual connections to mitigate overfitting. The experimental results show that both the best training loss and test loss still decrease slowly as the number of layers increases.

Furthermore, on Language Modeling tasks, we replaced 24 MLP Layers of Transformer with 24 FAN Layers, i.e. Transformer with FAN, and it also achieved clear improvements on each task, especially for OOD zero-shot evaluation scenarios. These findings indicate that FAN Layer is effective for deep neural networks.

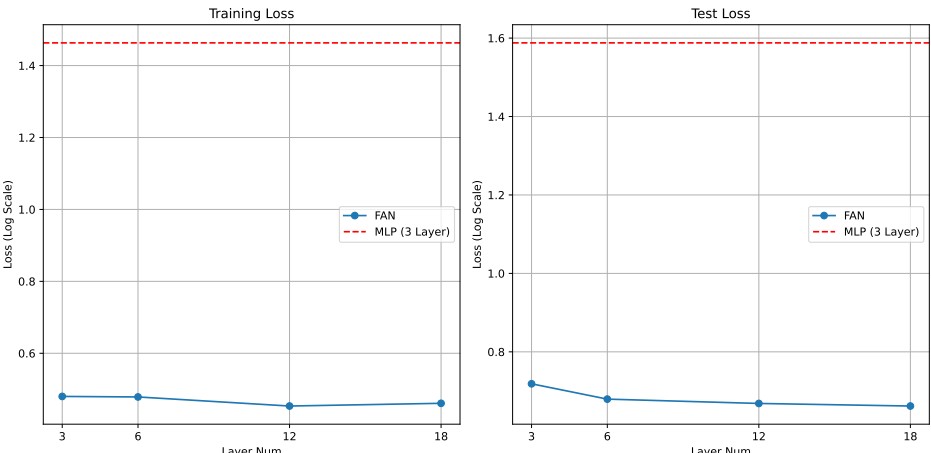

Figure 8: Performance of Deeper FAN on fitting $y = e^{\sin^2(\pi x) + \cos(x) + (x \mod 3)} - 1$.

## C EXPERIMENTAL DETAILS

### C.1 SETUP OF PERIODICITY MODELING

In periodicity modeling tasks, FAN, MLP, and KAN each consist of three layers with comparable FLOPs, while the Transformer model comprises twelve layers. For consistency, we set the hidden layer dimension ($d_h$) to 2048 for FAN, MLP, and Transformer. In the case of KAN, we follow its original paper (Liu et al., 2024), where the spline order ($K$) and the number of spline intervals ($G$) are set to 3 and 50, respectively. We apply a learning rate of $1 \times 10^{-5}$ for training all models. We ensured that the data density of each period in tasks was consistent, meaning that each cycle contained a fixed quantity of 10,000 training data points.

### C.2 SETUP OF SYMBOLIC FORMULA REPRESENTATION

In symbolic formula representation tasks, we used the create_dataset function from the official KAN repository to generate the datasets. Each dataset contains 3000 training samples and 1000 test samples, with all input variables randomly sampled from the range [-1, 1]. We followed the training settings from the original KAN paper, training all methods using LBFGS for 1800 steps. For KAN, we increased the number of grid points to scale up the parameter size, covering $G = \{3, 5, 10, 20, 50, 100, 200, 500, 1000\}$. For other methods, we scaled up the parameter size by increasing the number of layers and the dimensions of hidden layers.

### C.3 SETUP OF TIME SERIES FORECASTING

In time series forecasting task, we implement our model based on the codebase by (Wu et al., 2021b). Each model comprises 2 encoder layers and 1 decoder layer. We fix the hidden size for both the Transformer and our model at 512, with the feedforward dimension set to 2048 (four times the hidden size). The parameter sizes detailed in the main text correspond to the Exchange dataset;

variations in the number of variables across different datasets influence the linear layers in the model. We adjust the hidden sizes of the other models to align with the Transformer parameters for fairness.

## C.4 SETUP OF LANGUAGE MODELING

In language modeling task, we employ the BERT tokenizer (Devlin et al., 2018) and an embedding layer with a dimensionality of 768, except for Mamba, which adheres to its default settings as specified in the original paper (Gu & Dao, 2023). The architecture features 4, 24, and 12 layers with hidden sizes of 1800, 768, and 768 for LSTM, Mamba, and Transformers, respectively. To mitigate training stagnation in deeper LSTM models, we reduce the number of layers while increasing the hidden size to balance the parameters. Importantly, Mamba's layer count is twice that of a similarly sized Transformer, as each layer consists of two Mamba blocks (Multihead attention block + MLP block).

## C.5 SETUP OF IMAGE RECOGNITION

In image recognition tasks, we employ a simple CNN generated by ChatGPT as the baseline model, which consists of four Convolutional Layers and two MLP Layers. We replace MLP with FAN in CNN, i.e. CNN with FAN, as the counterpart, ensuring that they have similar parameters. For each task, we use stochastic gradient descent with momentum (SGDM) as the optimizer, the learning rate is set to 0.01, and the training process runs for 100 epochs.

## D COMPARISON OF FAN AND SNAKE ACTIVATION FUNCTION

We compare FAN with Snake, a previous approach used for improving the fitting of periodic functions with neural networks. The results are shown in Figure D.

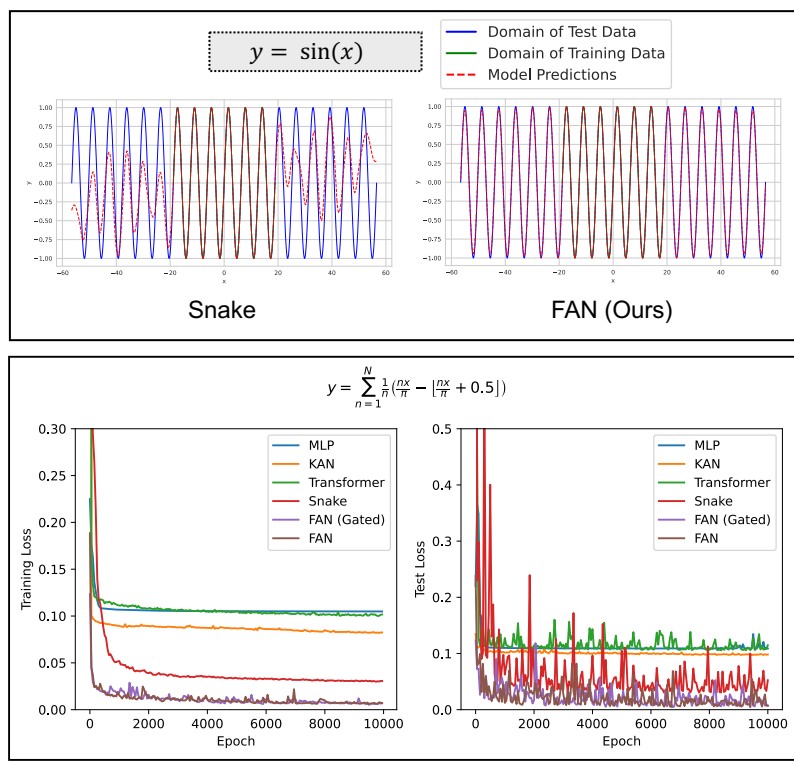

Figure 9: Comparisons of FAN with MLP (Snake) (Liu et al., 2020) in fitting periodic functions.

# E  HOW FAN COMPLY WITH UNIVERSAL APPROXIMATION THEOREM

The Universal Approximation Theorem states that a feed-forward network with a single hidden layer containing a finite number of neurons can approximate continuous functions on compact subsets of $\mathbb{R}^n$, under mild assumptions on the activation function, which needs to be a non-constant, no-linear, and continuous function. FAN Layer is defined as $\phi(x) = [\cos(W_p x)||\sin(W_p x)||\sigma(B_{\bar{p}} + W_{\bar{p}} x)]$, where $||$ denotes concatenation and $\sigma$ denotes the standard activation function, such as ReLU and GELU. Since $\sin$ and $\cos$ functions also satisfy the required conditions of being non-constant, continuous, and non-linear activation functions, the FAN layer adheres to the Universal Approximation Theorem.

# F  MORE DETAILS OF BASELINES

In our experiments, we mainly compare FAN with the following baselines. 1) **MLP** (Rosenblatt, 1958): the most classic model, which is widely used in the backbone of various models. 2) **Transformer** (Vaswani et al., 2017): a prevalent model known for its self-attention mechanism, which achieves outstanding performance on various tasks. 3) **KAN** (Liu et al., 2024): an emerged model specialized for symbolic formula representation, which uses the b-spline functions instead of fixed activation functions. 4) **LSTM** (Hochreiter & Schmidhuber, 1997): a well-known recurrent neural network (RNN) that can capture long-term dependencies on sequential data. 5) **Mamba** (Gu & Dao, 2023): an emerged selective state space model (SSM) that achieves competitive performance on some tasks with sequential inputs. 6) **CNN** (LeCun et al., 1998): convolutional neural network contains the convolutional layers, which are effective in processing image data.

