# OpenReview forum: "FAN: Fourier Analysis Networks"
_ICLR.cc/2025/Conference — Submitted to ICLR 2025_

### Official Review · Reviewer_D8kJ · 2024-10-19

**Soundness:** 3
**Presentation:** 3
**Contribution:** 2
**Rating:** 5
**Confidence:** 3

**Summary:**

+ This paper introduces Fourier Analysis Networks (FAN), a class of learning models with an inductive bias for periodic data.

+ The authors show that FAN match the performance of KAN and MLP, two similar architectures, on a variety of symbolic formula regression tasks.

+ The authors introduce a variety of baseline models, and they show that their method is competitive on general problems like time series forecasting and language modelling.

**Strengths:**

+ I think the paper is clearly written and makes a good-faith effort to benchmark, matching the design of benchmarks used in prior works like KAN. I appreciate that the authors benchmark on a wide variety of time series and language tasks. I particularly like the benchmarks on long-horizon forecasting datasets, like ETTh, which has a dominant periodic component that could potentially benefit from a model with an inductive bias for periodicity

+ The authors have included some interesting ablations and generalizations, including a Transformer architecture with FAN replacing the MLP layers, allowing them to directly process language data.

+ I very much appreciate that controlled for model size, by ensuring that all models shown in Table 2 are roughly the same size.

**Weaknesses:**

**Scope**

I read this paper as arguing that FAN represents a good general-purpose model for any task. The authors perform experiments showing that it exceeds the performance of KAN on general fitting tasks like fitting non-periodic symbolic functions or language modelling.
But, if this is really the case, why would this be true? What about the Fourier basis compared to the b-spline in KAN or sigmoids in MLP would allow it to perform better? I feel like this paper is missing either a theoretical justification, or a set of ablations or other experiments that probe the trained FAN models, that can help establish intuition for why this would work.

If the authors’ argument is instead that FAN has an inductive bias for periodic data, and so it should be best-in-class for a particular type of problem (like forecasting time series with mixed periodicity), then the experiments should be narrower and focus on establishing precedence on this problem. In this case, the model would need to be compared against simpler methods, including a simple Fourier transform forecast model (this is included in the darts library).

Either of the approaches above would make sense to me, but right now the paper seems to be arguing the need for a model with strong inductive biases for time series, and then doing experiments that seem to argue that this model is general-purpose,


**Claims**

+ I don’t completely buy that models like the Transformer can’t model purely periodic functions. Did the authors fairly tune the baselines here? Is this a known problem in the literature? This seems like it would be a widely-known and reported problem, which would preclude any prior progress on datasets like ETTh, and so I am skeptical of this claim. The papers the authors cite in support of this argument in the related work section ((Silvescu, 1999; Liu, 2013; Parascandolo et al., 2016; Uteuliyeva et al., 2020) only show periodic modifications to architectures; none make broader claims about periodic functions being inaccessible to learning models.

+ Why would this work on language data? The authors seem to be arguing that periodic functions are a more expressive general function class than the b-splines of KAN or linear -> activation structure of MLPs, since language itself has no periodic component.


**Novelty**

+ This is my primary justification for my score. I don’t necessarily think that modifying the architecture of KANs and MLP with a different transcendental function is conceptually novel, and the current text doesn’t provide any further mechanistic analysis of trained models

+ For a higher score I would either need a motivating application, perhaps a SciML problem where the Fourier function class is needed, or I would need some theoretical or empirical insight into why FAN works better than other methods as a general approximator.

**Questions:**

+ Why hasn't the issue with periodic functions been more widely studied? Are there prior works along these lines?

+ Why do the FAN Transformers work on the language task, given how different this is from the area where Fourier approaches are normally applied?

---

> ### Author Response · Authors · 2024-11-24
> **Response to Reviewer  D8kJ (Part 1/2)**
>
> Thanks very much for your comments. We will reply to your questions point-by-point below and look forward to having further discussions with you!
>
> > This is my primary justification for my score. I don’t necessarily think that modifying the architecture of KANs and MLP with a different transcendental function is conceptually novel ...
>
> We did not simply modify the activation function of KANs and MLP, and the architecture of FAN is different from them. Specifically,
>
> First, **we introduce Fourier principle into FAN's architecture.** All signals or tasks in the world can be divided into the superposition or combination of periodic and non-periodic parts. Therefore, enabling general neural networks to model periodicity is necessary. However, existing general neural networks struggle to model the periodicity from data, especially for OOD scenarios, indicating their potential flaws in the modeling and reasoning of periodicity. In this paper, by introducing Fourier principle, the periodicity is naturally integrated into the structure and computational processes of FAN, thus achieving a more accurate expression and prediction of periodic patterns.
>
> Second, **we address the problems of traditional Fourier Neural Networks**, i.e., approximate the Fourier coefficients are independent of the depth. We design FAN following the two key principles: 1) the capacity of FAN to represent the Fourier coefficients should be positively correlated to its depth, i.e., performing better as depth increases; 2) the output of any hidden layer can be employed to model periodicity using Fourier Series through the subsequent layers. Achieving both principles simultaneously is neither straightforward nor trivial. The first one enhances the expressive power of FAN for periodicity modeling by leveraging its depth, while the second one ensures that the features of FAN’s intermediate layers are available to perform periodicity modeling. As a result, FAN can serve as building blocks for deep neural networks.
>
> Third, **FAN has fewer parameters and FLOPs.** As shown in Table 1, FAN reduces the number of parameters and FLOPs by sharing the parameters and computation of Sin and Cos parts. Therefore, with the same input and output dimension, FAN has fewer parameters and FLOPs than MLP and KAN (KAN has more parameters and FLOPs than MLP with the same settings [1]), which also proves that FAN is not simply modifying the transcendental function.
>
> Fourth, **FAN has good performance on general tasks, especially for OOD scenarios.** Existing Fourier Neural Networks are usually designed as the specialized model for a single task. However, we demonstrated the effectiveness of FAN with extensive experiments on real-world applications, including Symbolic Formula Representation, Time Series Forecasting, Language Modeling, and Image Recognition tasks (updated in the revised version, see anonymous github for [detailed results](https://anonymous.4open.science/r/Additional-Experiments-of-FAN-DD10/README.md)).
>
> #### Table1: Comparison of MLP and FAN, where $d_\text{p}$ defaults to $\frac{1}{4}d_\text{output}$, $d_\text{input}$ and $d_\text{output}$ denote the input and output dimensions of layer. In our evaluation, FLOPs for any arithmetic operations are considered as 1, and for Boolean operations as 0.
> ||MLP Layer|FAN layer|
> |-|:-|:-|
> |Formula|$\Phi(x) = \sigma(B_{m} + W_{m}x)$|$\phi(x) = [\cos(W_px)\|\| \sin(W_px)\|\| \sigma(B_{\bar{p}} + W_{\bar{p}}x)]$|
> |Num of Params|$(d_\text{input} \times d_\text{output}) + d_\text{output}$ | $(1-\frac{d_p}{d_\text{output}})\times((d_\text{input} \times d_\text{output}) + d_\text{output})$|
> |FLOPs|$2\times(d_\text{input} \times d_\text{output})$\n  $+ d_\text{output} \times \text{FLOPs}_\text{non-linear}$|$(1-\frac{d_p}{d_\text{output}})\times(2\times(d_\text{input} \times d_\text{output}))$ $+ d_\text{output} \times \text{FLOPs}_\text{non-linear} $|
>
> Reference:
> [1] KAN or MLP: A Fairer Comparison.
>
> > For a higher score I would either need a motivating application, perhaps a SciML problem where the Fourier function class is needed ...
>
> We conduct experiments on the SciML problem that includes the Fourier function class following the work [2]. The Burgers’ equation, a non-linear partial differential equation, is frequently used in scientific computing to model shock waves and traffic flow, among other phenomena. The detailed error rate on Burgers’ equation is listed in the Table below, where the values represent the Average Relative Error and the lower values indicate better performance. We can find that replacing the MLP Layer with FAN Layer in Fourier Neural Operator (FNO) can achieve clear improvements on each setting of resolution *s* of this task.
>
> |Model|s=256|s=512|s=1024|s=2048|s=4096|s=8192|
> |-|-|-|-|-|-|-|
> |FNO|5.93%|6.14%|6.03%|6.75%|7.36%|9.93%|
> |FNO with FAN|**5.26%**|**5.17%**|**5.18%**|**6.73%**|**6.35%**|**7.06%**|
>
> Reference:
> [2] Fourier Neural Operator for Parametric Partial Differential Equations.

---

> ### Author Response · Authors · 2024-11-24
> **Response to Reviewer D8kJ (Part 2/2)**
>
> > Why hasn't the issue with periodic functions been more widely studied? Are there prior works along these lines? (I don’t completely buy that models like the Transformer can’t model purely periodic functions. Is this a known problem in the literature?...)
>
> The issue of models like transformer having potential flaws in periodicity modeling has already been discussed. Some works also pointed to this argument, which were cited in our Related Work Section. For examples, Works [3] and [4] pointed out that Transformers still struggle with modeling periodicity. Work [5] confirmed that deep neural networks fail to learn purely periodic functions, as demonstrated in its Figure 1, and introduced the Snake function, i.e., \(x + \sin^2(x)\), as the activation function to address this issue. However, we observed that it can fit periodic functions to a certain extent, but its effect is limited especially for OOD scenarios, as demonstrated in Appendix D. Therefore, although some previous studies have discussed this issue, their actual performance and range of applications remain heavily constrained.
>
> In this paper, we focus on addressing the issues that existing general neural networks struggle to model the periodicity from data, especially for OOD scenarios. Our proposed FAN performs exceptionally well on periodicity modeling and achieves competitive performance on a range of general real-world tasks.
>
> Reference:
> [3] Conditional Generation of Periodic Signals with Fourier-Based Decoder. NeurIPS
> [4] Bridging Self-Attention and Time Series Decomposition for Periodic Forecasting. CIKM
> [5] Neural Networks Fail to Learn Periodic Functions and How to Fix It. NeurIPS
>
> > Why do the FAN Transformers work on the language task ...
>
> Many machine learning tasks, such as language modeling, may harbor hidden forms of periodicity and require periodicity modeling for OOD scenarios. For instance, consider addition and multiplication operations: despite being strictly non-periodic functions, they involve bit-level operations and carry mechanisms that exhibit periodic characteristics. Moreover, from language perspective, periodicity is not just a data feature but reflects a form of structural knowledge—one that allows for the transfer and reuse of abstract rules and principles across different contexts. Therefore, FAN’s ability to capture the latent periodic features contributes to its enhanced performance in these tasks to some extent. The conclusion stems from our experiments, where using FAN exhibits clear improvement especially for OOD scenarios in extensive experiments on Symbolic Formula Representation, Time Series Forecasting, Language Modeling, and Image Recognition tasks.
>
> > ... the performance of KAN on general fitting tasks like fitting non-periodic symbolic functions or language modelling. But, if this is really the case, why would this be true? ...
>
> We didn't compare our approach with KAN on Language Modeling tasks, since  KAN's paper only conducted experiments on Symbolic Formula Representation tasks, and work [1] claims that "except for symbolic formula representation tasks, MLP generally outperforms KAN."
>
> In Symbolic Formula Representation tasks, when the parameters are small, our approach achieves competitive results with KAN. However, when the parameters are large, our approach significantly outperforms KAN, as KAN shows a collapse and is even worse than MLP. In fact, the observation of KAN's performance collapse at large parameters is also described on Page 10 of KAN's paper, "... the test losses first go down then go up, displaying a U-shape, due to the bias-variance tradeoff ..." Figure 2.3 of KAN's paper also shows this phenomenon as the grid increases when fitting $f(x, y) = exp(sin(x)+y^2)$, which is included in our experiments as well. Moreover, as we know, some works also observe similar experimental results, such as Figure 8 of work [6] and Figure 6 of work [7].
>
> Reference:
> [1] KAN or MLP: A Fairer Comparison.
> [6] Kolmogorov Arnold Informed neural network: A physics-informed deep learning framework for solving PDEs based on Kolmogorov Arnold Networks.
> [7] On Training of Kolmogorov-Arnold Networks.

---

> > ### Author Response · Authors · 2024-11-26
> > **Looking forward to your feedback**
> >
> > Dear Reviewer D8kJ,
> >
> > We appreciate all of the valuable time and effort you have spent reviewing our paper. As today is the last day of the discussion period, we gently request that you review our reply and consider updating your evaluation accordingly. We believe that we have addressed all questions and concerns raised, but please feel free to ask any clarifying questions you might have before the end of the discussion period.
> >
> > Best,
> > Authors

---

> > > ### Comment · Reviewer_D8kJ · 2024-11-26
> > >
> > > Thank you for your response. I appreciate the references to the limitations of neural networks in learning periodic functions, which  is an interesting problem in the literature that I have not encountered previously.
> > >
> > > I appreciate the authors' reply, and I can see how this approach can have value, but **the current experiments are not sufficient to convince me that this approach has the novelty or performance needed for me to assign a higher score.**
> > >
> > > I would suggest that, in future work, the authors keep the existing motivation (NNs can't learn periodic functions), and then segue into a more specific problem where this periodic limitation leads to clear failure of existing approaches. The Burgers' equation results with FNOs the authors mention in the rebuttal period would be a great motivating example, as would a more extensive time series problem with a periodic component, like physiological data or EEGs (current forecast models still underperform in this domain, perhaps due to the periodicity issue identified by the authors). The language modeling example still doesn't seem well-justified to me, and would be better suited for an appendix.

---

> ### Author Response · Authors · 2024-11-30
>
> Dear Reviewer D8kJ,
>
> Thanks for your suggestions. **We are glad that you can see the value of FAN.** However, we also would like to clarify that **we conducted extensive experiments to verify the effectiveness of FAN**, specifically: 1) We verified the powerful periodicity modeling capabilities of FAN across various periodicity modeling tasks, as shown in Figure 1, Figure 3, Figure 6, and Figure 9. 2) We verified the effectiveness and generalizability of FAN in extensive real-world tasks, including symbolic formula representation, time series forecasting, language modeling, and image recognition. 3) We have added the Further Analysis of FAN in Section 4.6, including the impact of hyperparameter $d_p$ and the runtime of FAN, providing deeper insights into its efficiency. 4) We have also updated additional experiments in Appendix B.1 to B.7, e.g., the SciML problem you mentioned and further experiments on time series problems.
>
> Regarding time series forecasting tasks, the most frequently studied public datasets, such as the Weather, Exchange, Traffic, and ETTh datasets [8, 9, 10], have been employed in our experiments, and to the best of our knowledge,  the related works didn't conduct experiments on physiological data or EEGs. For physiological data or EEGs, can you specify some public benchmark or related work of time series forecasting that conducts experiments on physiological data or EEGs?
>
> Finally, to address the poor extrapolation of these general neural networks such as MLP and Transformer in periodicity modeling, our work designs a general model building block. For general real-world tasks (like language modeling and image recognition), using FAN achieved competitive or superior performance than the baselines, **especially for dealing with OOD scenarios**, since the real-world tasks inherently contain many periodic and non-periodic features, although some of them are hidden.
>
> We look forward to your feedback and further discussions with you.
>
> Sincerely,
> Authors
>
> Reference:
> [8] Autoformer: Decomposition transformers with auto-correlation for long-term series forecasting. NeurIPS
> [9] Informer: Beyond efficient transformer for long sequence time-series forecasting. AAAI
> [10] FEDformer: Frequency Enhanced Decomposed Transformer for Long-term Series Forecasting. ICML

---

### Official Review · Reviewer_RaTJ · 2024-10-26

**Soundness:** 1
**Presentation:** 2
**Contribution:** 2
**Rating:** 3
**Confidence:** 3

**Summary:**

This paper introduces FAN (Fourier Analysis Network), a new neural network architecture designed to better model periodic patterns. Unlike traditional neural networks, which often struggle to generalize periodic data and tend to memorize it instead, FAN uses Fourier Series to directly incorporate periodicity into the model’s structure. This approach allows FAN to more effectively learn and generalize periodic functions, especially in cases where the data goes beyond the training range. The authors show that FAN can replace multi-layer perceptrons (MLPs) in many tasks, achieving similar or better results with fewer parameters and computational needs. Experiments demonstrate FAN’s strengths across tasks like time series forecasting, symbolic formula representation, and language modeling. FAN outperforms models such as MLPs, KAN, and Transformers, highlighting its effectiveness in both periodic and non-periodic contexts. Additionally, FAN achieves competitive or superior results in real-world applications, showing potential as a foundational model component.

**Strengths:**

The paper is well-written, and I was able to quickly understand both the core concepts and the practical implementation. The numerous graphs effectively illustrate the methodology and results, enhancing the clarity and impact of the work.

**Weaknesses:**

1.	When discussing the advantages of predicting Fourier coefficients, consider citing the theorem that emphasizes the significance of Fourier coefficient singularity. Additionally, referencing the foundational property that smooth functions can often be approximated well with a finite number of Fourier coefficients would strengthen the argument that predicting the first ￼ coefficients is typically sufficient.
	2.	The syntax and structure are somewhat confusing. Specifically, the distinction between ￼ and ￼ is unclear upon initial reading and could benefit from a more explicit explanation.
	3.	In demonstrating results on known functions, it is worth noting that functions with a finite number of Fourier coefficients (finite in the frequency domain but infinite in the time domain) will predictably perform better when analyzed in the frequency domain. Consequently, comparing this to models that do not use the frequency domain is not particularly informative. Instead, consider comparing with frequency-based models such as FRETS, FEDformer, and other similar approaches. Furthermore, adding examples with different types of functions would provide a more comprehensive evaluation, such as:
	•	Periodic functions that include both low and high frequencies (e.g., ￼).
	•	Non-periodic functions.
	4.	When discussing other real-world datasets, the choice of datasets could be expanded. The omission of non-periodic datasets, such as exchange rates and electricity usage, is particularly noticeable, and including them would enhance the comprehensiveness of your analysis.
	5.	In the results section, comparing your model solely with non-frequency models may be less meaningful, as the use of FFT in time series prediction is not a novel approach, and several previous models have integrated FFT with MLPs. A more relevant comparison would be to other FFT-based models to effectively demonstrate the advantage of your approach, particularly in terms of the integration between FFT coefficients and linear modeling.
	6.	If you are presenting the FLOP count, it would be beneficial to include a runtime graph. My intuition suggests that the MLP model may have a higher runtime, and a runtime graph would provide a clearer perspective on computational efficiency.
	7.	Including a link to the code within the abstract would be valuable for readers interested in exploring the implementation.

**Questions:**

1.	Clarify the distinction between your approach and directly learning the coefficients.
	•	Suggestion: Conduct an ablation study to quantify the improvement in results when incorporating the linear component.
	2.	Explain the effect of the parameter ￼ on model performance.
	•	Recommendation: Provide details on how ￼ influences the results.

---

> ### Author Response · Authors · 2024-11-19
> **Response to Reviewer RaTJ (Part 1/2)**
>
> All of the mathematical formulas in your comment are garbled, significantly impairing the readability and comprehension. For examples, "...the distinction between ￼ and ￼ is unclear...", "... include both low and high frequencies (e.g., ￼)", "Explain the effect of the parameter ￼ on model performance.", and "Provide details on how ￼ influences the results.". Please correct these mathematical formulas to enhance the clarity. We will try our best to answer your questions below. Moreover, some of the questions you raised were already addressed in our paper, which you may miss, and we will restate them. We look forward to further discussions with you.
>
> > Comparing with frequency-based (FFT-based) models in the Time Series Forecasting tasks.
>
> We did not compare with these specialized models because we aim to design a general model building block, which addresses the poor extrapolation of these general neural networks such as MLP and Transformer in periodicity modeling. This is also the reason we conduct the extensive experiments on Symbolic Formula Representation, Time Series Forecasting, Language Modeling tasks, and Image Recognition tasks (updated in the revised version, see anonymous github for [detailed results](https://anonymous.4open.science/r/Additional-Experiments-of-FAN-DD10/README.md)) to prove the effectiveness of FAN. These specialized models are hard to cover all these tasks.
>
> For the frequency-based models in Time Series Forecasting tasks, we recommend you employ them simultaneously, i.e., replace MLP with FAN in frequency-based models, to achieve better results. We present the experimental results below, where the results of FEDformer are cited from its paper directly. From the results, we can find that FEDformer with FAN can outperform FEDformer in almost all cases.
>
> |Dataset|Len|FEDformer||with FAN||
> |-|-|-|-|-|-|
> |||MSE|MAE|MSE|MAE|
> |Traffic|96|0.587|0.366|**0.577**|**0.357**|
> ||192|0.604|0.373|**0.601**|**0.366**|
> ||336|0.621|0.383|**0.620**|**0.378**|
> ||720|0.626|0.382|**0.619**|**0.370**|
> |Exchange|96|0.148|0.278|**0.138**|**0.267**|
> ||192|0.271|0.380|**0.261**|**0.371**|
> ||336|**0.460**|**0.500**|0.461|0.503|
> ||720|1.195|0.841|**1.159**|**0.827**|
> |Electricity|96|0.193|0.308|**0.184**|**0.298**|
> ||192|0.201|0.315|**0.199**|**0.313**|
> ||336|0.214|0.329|**0.212**|**0.325**|
> ||720|0.246|0.355|**0.239**|**0.347**|
>
>
> > Clarify the distinction between your approach and directly learning the coefficients.
>
> Without well-definition of "Directly learning the coefficients", we find it difficult to respond to this question. Our FAN Layer models the Fourier coefficients with the following principles: 1) the capacity of FAN to represent the Fourier coefficients should be positively correlated to its depth; 2) the output of any hidden layer can be employed to model periodicity using Fourier Series through the subsequent layers. The first one enhances the expressive power of FAN for periodicity modeling by leveraging its depth, while the second one ensures that the features of FAN’s intermediate layers are available to perform periodicity modeling.
>
> Does "Directly learning the coefficients" in your opinion mean inputting sin(x) and cos(x) and then using MLP Layer instead of FAN Layer to model the Fourier coefficients? In this setting, frequencies are fixed and only the coefficients are learned, which may limit the model's ability to capture patterns not aligned with these frequencies. Taking simple f(x) = x mod 5 as an example, this setting may not even converge at all, because the frequency of x mod 5 is inconsistent with sin(x) and cos(x). The experimental results of their loss are shown as follows.
>
> |Epoch|50|100|150|200|
> |-|:-|:-|:-|:-|
> |Directly learning the coefficients|2.10|2.09|2.09|2.08|
> |FAN|0.28|0.23|0.18|0.17|
>
> > Conduct an ablation study to quantify the improvement in results when incorporating the linear component.
>
> We disagree. Do you mean removing the linear component in sin and cos part, i.e., changing the expression from $[\cos(W_px)\|\| \sin(W_px)\|\| \sigma(B_{\bar{p}} + W_{\bar{p}}x)]$ to $[\cos(x)\|\| \sin(x)\|\| \sigma(B_{\bar{p}} + W_{\bar{p}}x)]$? We are uncertain why we need to compare with it, as it is obviously a specific case of our approach. If $W_p$ is the identity matrix, then the representational effects of both approaches become equivalent. However, our approach offers the flexibility to learn more customized combinations of the previous layer’s features. Moreover, there appears to be a potential issue with dimension alignment. This modification would make the output dimension at least twice the input dimension, leading to exponential growth in dimensions with increased network depth.
>
> The reason we didn't conduct the ablation study was that the operations within our FAN Layer are integrated as a whole. Removing any one of these operations would violate the above-mentioned principles and cause the entire network not to work normally.

---

> > ### Author Response · Authors · 2024-11-19
> > **Response to Reviewer RaTJ (Part 2/2)**
> >
> > > Adding examples with different types of functions would provide a more comprehensive evaluation, such as: Periodic functions that include both low and high frequencies. Non-periodic functions.
> >
> > We already included them in our experiments and you can find them in Figure 4 and Figure 5.
> >
> > > The omission of non-periodic datasets, such as exchange rates and electricity usage, is particularly noticeable, and including them would enhance the comprehensiveness of your analysis.
> >
> > We already included your mentioned non-periodic dataset, i.e., exchange, in our experiments. In the exchange dataset, using FAN exhibits the relative improvements of 14.2% in MSE and 6.7% in MAE.
> >
> > > If you are presenting the FLOP count, it would be beneficial to include a runtime graph.
> >
> > Thanks for your suggestion. We present the experimental results below, which will be added to our revised manuscript. We can find that due to PyTorch's optimization of MLPs, they exhibit smaller runtimes when the input and output sizes are small. However, as the input and output sizes continue to increase, matrix computations become the main contributor to runtime. At this point, FAN's fewer parameters and reduced FLOPs begin to show significant advantages. Note that FAN can be further optimized from the underlying implementation, we leave this to future research.
> >
> > ||1024$\times$1024|2048$\times$2048|4096$\times$4096|8192$\times$8192|
> > |-|:-|:-|:-|:-|
> > |MLP|**0.064 ms**|**0.114 ms**|0.212 ms|0.938 ms|
> > |FAN|0.128 ms|0.133 ms|**0.211 ms**|**0.704 ms**|
> >
> > > Including a link to the code within the abstract would be valuable for readers interested in exploring the implementation.
> >
> > Taking into account the anonymity required during the review process, we provided the code of this paper in Supplementary Material, so you can directly find it if you want. If the paper is accepted, we will remove the anonymity of the code link and include it in the abstract.

---

> > > ### Author Response · Authors · 2024-11-21
> > > **Looking forward to your feedback**
> > >
> > > Hello, do our responses address your questions? Please let us know if there are any other questions you'd like to discuss!

---

> > > > ### Comment · Reviewer_RaTJ · 2024-12-02
> > > > **feedback**
> > > >
> > > > I appreciate the authors' response and can recognize the potential value of this approach. However, the existing experiments do not sufficiently demonstrate the novelty or performance required for me to assign a higher score.

---

> ### Author Response · Authors · 2024-12-03
>
> Thank you for your comments. **We appreciate that you recognize the FAN's value.** The experiments you mentioned have been conducted in our paper or explain the unreasonability in the response. **Could you explain what aspect of experiments you require to further demonstrate the novelty or performance of FAN?** (We would also be grateful if you could correct the garbled mathematical formulas of your original responses to help us understand.) We would like to clarify that we conducted extensive experiments to verify the generalizability and effectiveness of FAN, specifically:
>
> * We verified the powerful periodicity modeling capabilities of FAN across various periodicity modeling tasks, as shown in Figure 1, Figure 3, Figure 6, and Figure 9.
> * We verified the effectiveness and generalizability of FAN in extensive real-world tasks, including symbolic formula representation, time series forecasting, language modeling, and image recognition.
> * We have added the Further Analysis of FAN in Section 4.6, including the impact of hyperparameter $d_p$ and the runtime of FAN, providing deeper insights into its efficiency.
> * We have updated the additional experiments in Appendix B.1-B.7, including further experiments on time series problems, FAN for solving SciML problems, the effectiveness of FAN Layer for deep neural networks, and so on.
>
> We look forward to your feedback and further discussions with you.

---

### Official Review · Reviewer_DFRd · 2024-10-31

**Soundness:** 3
**Presentation:** 3
**Contribution:** 3
**Rating:** 6
**Confidence:** 3

**Summary:**

This paper proposes a new network architecture FAN based on Fourier analysis, which improves the ability of efficient modeling and reasoning of periodic phenomena. By introducing Fourier series, periodicity is naturally integrated into the structure and calculation process of neural network, and more accurate expression and prediction of periodicity pattern are realized. FAN can seamlessly replace MLP in a variety of models while using fewer parameters and FLOPs. Through numerous experiments, the paper demonstrates the effectiveness of FAN in modeling and reasoning periodic functions, as well as its superiority and generalization ability for a range of real world tasks including symbolic formula representation, time series prediction, and language modeling.

**Strengths:**

1. This paper cleverly integrates Fourier analysis into neural network architecture, including both periodic and aperiodic parts, which can seamlessly replace MLP layer.
2. FAN uses fewer parameters and FLOPs than traditional MLPS and is more computationally efficient.
3. This model is more interpretable than the traditional black box model, such as MLP.

**Weaknesses:**

1. FLOPs analysis is given in this paper, but the comparison of actual training time is lacking
2. Will the result of a deeper FAN yield a better fit? Will overfitting results be produced? It is recommended to test the performance changes of layers 3-20 at different depths
3. This article compares performance with traditional MLP, KAN and transformer. Whether to compare and discuss with deep learning frameworks that actually deal with time series data, such as LSTM, RNN

**Questions:**

1. The paper lacks theoretical justification for why FAN enhances generalization capability;
2. They don't explain how FAN avoids overfitting, such as through some model selection methods.

---

> ### Author Response · Authors · 2024-11-24
> **Response to Reviewer DFRd**
>
> Thanks very much for your comments. We will reply to your questions point-by-point below and look forward to having further discussions with you!
>
> > The paper lacks theoretical justification for why FAN enhances generalization capability.
>
> From a qualitative perspective, real-world tasks inherently contain many periodic and non-periodic features (although some of these features are hidden). However, existing general neural networks struggle to model the periodicity from data, especially for OOD scenarios, indicating their potential flaws in the modeling and reasoning of periodicity. By introducing Fourier principle, FAN addresses the aforementioned challenges and has the ability to capture and model both periodic and non-periodic features, so FAN can enhance generalization capability. We provided this analysis in our Discussion Section.
>
> In this paper, we provide extensive experiments to demonstrate the generalizability of FAN, which achieves clear improvements on a range of real-world applications, including Symbolic Formula Representation, Time Series Forecasting, Language Modeling, and Image Recognition tasks (updated in the revised version, see anonymous github for [detailed results](https://anonymous.4open.science/r/Additional-Experiments-of-FAN-DD10/README.md)).
>
> > They don't explain how FAN avoids overfitting, such as through some model selection methods.
>
> We didn't adopt any specific methods to prevent overfitting for FAN. All settings are consistent for FAN and the baseline, i.e., we train them for a predetermined number of epochs. In Figure 4, we also displayed the training and testing loss of different models throughout the process of learning complex periodic functions. From Figure 4, we can observe that the performance of FAN significantly outperforms other baselines across all epochs, both in training and testing loss.
>
> > FLOPs analysis is given in this paper, but the comparison of actual training time is lacking
>
> Due to PyTorch’s optimization of MLP, although FAN has lower FLOPs, the actual training time of replacing MLP with FAN is similar. We conduct another experiment to show the advantages of FAN's lower FLOPs below, which will be added to our revised manuscript. We can find that MLPs exhibit smaller runtimes when the input and output sizes are small, due to PyTorch’s optimization of MLP. However, as the input and output sizes continue to increase, matrix computations become the main contributor to runtime. At this point, FAN's fewer parameters and reduced FLOPs begin to show significant advantages. Note that FAN can be further optimized from the underlying implementation, we leave this to future research.
>
> ||1024$\times$1024|2048$\times$2048|4096$\times$4096|8192$\times$8192|
> |-|:-|:-|:-|:-|
> |MLP|**0.064 ms**|**0.114 ms**|0.212 ms|0.938 ms|
> |FAN|0.128 ms|0.133 ms|**0.211 ms**|**0.704 ms**|
>
> > Will the result of a deeper FAN yield a better fit? Will overfitting results be produced? It is recommended to test the performance changes of layers 3-20 at different depths.
>
> Yes, a deeper FAN usually yields a better fit in our experiments.  We have evaluated the effect of varying the number of FAN layers from 3 to 20 on periodicity modeling tasks, employing residual connections to mitigate overfitting, as shown in Figure 8 of Appendix. The experimental results show that both the best training loss and test loss still decrease slowly as the number of layers increases.
>
> Furthermore, on Language Modeling tasks, we replaced 24 MLP Layers of Transformer with 24 FAN Layers, i.e. Transformer with FAN, and it also achieved clear improvements on each task, especially for OOD zero-shot evaluation scenarios. These findings indicate that FAN Layer is effective for deep neural networks.
>
> > This article compares performance with traditional MLP, KAN and transformer. Whether to compare and discuss with deep learning frameworks that actually deal with time series data, such as LSTM, RNN
>
> We compared with LSTM (a well-known RNN) on Time Series Forecasting and Language Modeling tasks, as demonstrated in Tables 2 and 3, respectively.

---

### Official Review · Reviewer_7Gdo · 2024-11-03

**Soundness:** 2
**Presentation:** 2
**Contribution:** 1
**Rating:** 3
**Confidence:** 5

**Summary:**

The paper addresses the critical challenge of periodicity modeling in neural networks, proposing the Fourier Analysis Network (FAN) to overcome inherent limitations of MLPs and Transformers in handling periodic data. The attempt to leverage Fourier Series for improving periodicity modeling is a novel approach, and integrating this into neural network design could potentially enhance tasks like time series forecasting and symbolic formula representation.

**Strengths:**

The paper proposes a novel approach based on Fourier Analysis, which is a well-established mathematical tool for modeling periodic functions. The integration of Fourier Series into the network design could potentially address the limitations of traditional architectures like MLP and Transformer in periodicity modeling.

The writing is generally clear and the figures, such as Figure 1 and others, are well-presented. They help in visualizing the performance of FAN relative to other models.

The experiments cover a variety of tasks, including symbolic formula representation, time series forecasting, and language modeling, which demonstrates the general applicability of the proposed architecture.

**Weaknesses:**

Insufficient Theoretical Justification:
The paper lacks a rigorous theoretical foundation that convincingly demonstrates the superiority of FAN over existing methods. While the integration of Fourier Series is discussed, the connection to the universal approximation theorem and how FAN mitigates spectral bias needs clearer elaboration.

Inadequate Experimental Validation:
The experiments provided are not sufficient to substantiate the claims. The datasets used for evaluation, especially in time series forecasting and symbolic formula representation, do not comprehensively cover real-world complexities. Additionally, key experimental details such as hyperparameter tuning, model initialization, and evaluation metrics are missing or inadequately described.

Limited Comparisons with Baselines:
The paper claims FAN outperforms MLPs, KANs, and Transformers, yet the comparisons lack depth. There is insufficient analysis of why FAN performs better, and the results for non-periodic tasks suggest that FAN’s benefits might be overstated.  It lacks comparisons with other neural networks that also leverage Fourier analysis. The absence of this comparison makes it difficult to fully evaluate the effectiveness of FAN, and this omission raises questions about the extent of its claimed advantages.

Unclear Practical Utility:
While the theoretical motivation is apparent, the practical implications are less clear. The paper does not provide compelling use cases or applications where FAN’s periodicity modeling significantly outperforms traditional methods in practical scenarios.

**Questions:**

1. Lack of detailed explanation about parameter efficiency:
The abstract claims that FAN can substitute MLP with fewer parameters. However, the theoretical justification and experimental evidence supporting this claim are not sufficiently detailed in the main text. It would be beneficial to include a comprehensive analysis comparing the parameter count and FLOPs of FAN versus MLP in various tasks.

2. Doubtful results in symbolic formula representation:
In Figure 5, the performance of KAN (Liu et al., 2024) was not replicated accurately. As KAN has shown strong results in symbolic formula representation in previous works, the omission of this accuracy raises concerns. I suggest the authors provide a more in-depth discussion of why FAN outperforms KAN in their experiments, or at least explain the differences in experimental settings that may have led to this discrepancy.

3. Practical Use Cases:
Can the practical utility of Fourier Approximation Networks (FAN) be demonstrated with real-world applications beyond synthetic and controlled experiments? How does FAN improve performance on industry-relevant tasks?

4. Clarification of Contributions:
What aspects of FAN are novel compared to existing Fourier-based neural networks? Can the authors clarify potential overlaps with prior work and better situate FAN within the context of ongoing research?

---

> ### Author Response · Authors · 2024-11-19
> **Response to Reviewer 7Gdo (Part 1/3)**
>
> **Most of your questions have been addressed in our original paper, and we will restate these arguments.** We reply to all your questions point-by-point below, hoping they can answer your questions, and look forward to your high-quality replies and following discussions with you.
>
> > 1.Lack of detailed explanation about parameter efficiency: The abstract claims that FAN can substitute MLP with fewer parameters. However, the theoretical justification and experimental evidence supporting this claim are not sufficiently detailed in the main text. ...
>
> We disagree. We already demonstrated the parameter efficiency of FAN in our original paper through both theoretical analysis and experimental results. In theoretical analysis, FAN reduces the number of parameters and FLOPs by sharing the parameters and computation of Sin and Cos parts. As shown in Table 1 of our paper, we provided the theoretical differences in both parameters and FLOPs, and their main difference is the preceding coefficient (i.e. $(1-\frac{d_p}{d_\text{output}}) < 1$ ) in FAN Layer, so FAN can substitute MLP with fewer parameters and FLOPs. Moreover, in our experiments, we aslo presented the comparisons of parameter counts in various tasks, such as Time Series Forecasting tasks (Table 2), Language Modeling tasks (Table 3), and Symbolic Formula Representation tasks (Figure 5).
>
> #### Table1: Comparison of MLP layer and FAN layer, where $d_\text{p}$ is a hyperparameter of FAN layer and defaults to $\frac{1}{4}d_\text{output}$ in this paper, $d_\text{input}$ and $d_\text{output}$ denote the input and output dimensions of the neural network layer, respectively. In our evaluation, the FLOPs for any arithmetic operations are considered as 1, and for Boolean operations as 0.
> ||MLP Layer|FAN layer|
> |-|:-|:-|
> |Formula|$\Phi(x) = \sigma(B_{m} + W_{m}x)$|$\phi(x) = [\cos(W_px)\|\| \sin(W_px)\|\| \sigma(B_{\bar{p}} + W_{\bar{p}}x)]$|
> |Num of Params|$(d_\text{input} \times d_\text{output}) + d_\text{output}$ | $(1-\frac{d_p}{d_\text{output}})\times((d_\text{input} \times d_\text{output}) + d_\text{output})$|
> |FLOPs|$2\times(d_\text{input} \times d_\text{output})$  $+ d_\text{output} \times \text{FLOPs}_\text{non-linear}$|$(1-\frac{d_p}{d_\text{output}})\times(2\times(d_\text{input} \times d_\text{output}))$ $+ d_\text{output} \times \text{FLOPs}_\text{non-linear} $|
>
> > 2.Doubtful results in symbolic formula representation: In Figure 5, the performance of KAN was not replicated accurately. As KAN has shown strong results in symbolic formula representation in previous works, the omission of this accuracy raises concerns. ...
>
> We wholeheartedly disagree. **The experimental results in symbolic formula representation are correct and reproducible, as we directly inherited the code from KAN's project, and our results are consistent with the observations made in KAN as well as some related studies**，specifically:
>
> As stated in our first paragraph of Symbolic Formula Representation Section, "We follow the experiments conducted in KAN’s paper, adhering to the same tasks, data, hyperparameters, and baselines. In addition to the original baselines, we also include Transformer for comparison in this task." We directly inherited the code from the KAN project for our experiments, and all results are reproducible using the code we provided. We have further thoroughly verified the code and confirmed there is no problem within it.
>
> In Figure 5, we compare our approach with the baselines across varying numbers of parameters on symbolic formula representation tasks. Following KAN's paper, the parameter size of KAN is increased by adding grid points $G$ with a fixed architecture they provided (Details can be found on Page 14 of KAN's paper), and for other baselines, the parameter size of the i-th layer baseline is increased by adding hidden size.
>
> Comparing the experimental results—Figure 5 in our paper and Figure 3.1 in KAN’s paper—you can find that all baseline results are consistent. Figure 3.1 only provided KAN's partial results in the range of small parameters, and complete results for the remaining baselines. In Figure 5 of our paper, we presented the complete results of all approaches.
>
> **In fact, the observation of KAN's performance collapse at large parameters is also described on Page 10 of KAN's paper, "... the test losses first go down then go up, displaying a U-shape, due to the bias-variance tradeoff ..." Figure 2.3 of KAN's paper also shows this phenomenon as the grid increases when fitting $f(x, y) = exp(sin(x)+y^2)$, which is included in our experiments as well.** Moreover, as we know, some works also observe similar experimental results, such as Figure 8 of work [1] and Figure 6 of work [2].
>
> Reference:
> [1] Kolmogorov Arnold Informed neural network: A physics-informed deep learning framework for solving PDEs based on Kolmogorov Arnold Networks.
> [2] On Training of Kolmogorov-Arnold Networks.

---

> > ### Author Response · Authors · 2024-11-19
> > **Response to Reviewer 7Gdo (Part 2/3)**
> >
> > > 3.Practical Use Cases: Can the practical utility of Fourier Approximation Networks (FAN) be demonstrated with real-world applications beyond synthetic and controlled experiments? ...
> >
> > First of all, our approach is "Fourier Analysis Network", which is based on Fourier analysis, NOT the so-called undefined network, "Fourier Approximation Networks" in your question.
> > Second, we already conducted extensive experiments on real-world applications, such as Time Series Forecasting on Weather, Exchange, Traffic, ETTh datasets and Language Modeling on SST-2, IMDB, Sentiment140, Amazon Reviews datasets that use raw data from real-world environments. We find that using FAN achieves clear improvements for Time Series Forecasting tasks, which have obvious periodic features, and also demonstrates better performance in language modeling tasks, which have latent or implicit periodic features, on cross-domain zero-shot evaluations.
> >
> > > 4.Clarification of Contributions: What aspects of FAN are novel compared to existing Fourier-based neural networks? Can the authors clarify potential overlaps with prior work and better situate FAN within the context of ongoing research?
> >
> > Existing Fourier-based neural networks to approximate the Fourier coefficients are independent of the depth, while the capacity of FAN to represent the Fourier coefficients is positively correlated to its depth, i.e., performing better as depth increases, therefore FAN can serve as building blocks for deep neural networks. This part is emphasized in our second paragraph of Related Work Section, i.e., Existing approaches "generally possess a similar principle as Eq. (3), using a neural network to simulate the formula of Fourier Series. However, this leads to the same problem as in Eq. (5), i.e., they are hard to serve as building blocks for deep neural networks, which limits these approaches' capabilities. In this paper, we design FAN to address these challenges, which performs exceptionally well on periodicity modeling and a range of real-world tasks." We detailedly elaborate on their problems and how we address these challenges on Page 3 of our paper.
> >
> > We also discussed the direction of Learning Periodicity with Neural Networks in Related Work Section, which partially overlaps with our work, i.e., "Commonly used neural networks, such as MLPs and transformers, struggle to extrapolate periodic functions beyond the scope of the training data. This limitation arises from the lack of inherent ``periodicity" in their inductive biases. Some previous works (Silvescu, 1999; Liu, 2013; Parascandolo et al., 2016; Uteuliyeva et al., 2020) proposed merely using standard periodic functions themselves or their linear combinations as activation functions, which only work well on some shallow and simple models. On this basis, work (Liu et al., 2020) introduced the Snake function, i.e., x + sin2 (x), as the activation function. However, we observed that it can fit periodic functions to a certain extent, but its effect is limited, as demonstrated in Appendix D. Therefore, although some previous studies have attempted to integrate the periodic information into neural networks, their actual performance and range of applications remain heavily constrained."
> >
> > > The connection to the universal approximation theorem and how FAN mitigates spectral bias needs clearer elaboration.
> >
> > The Universal Approximation Theorem states that a feed-forward network with a single hidden layer containing a finite number of neurons can approximate continuous functions on compact subsets of $\mathbb{R}^n$, under mild assumptions on the activation function, which needs to be a non-constant, no-linear, and continuous function. FAN Layer is defined as $\phi(x) = [\cos(W_px)\|\| \sin(W_px)\|\| \sigma(B_{\bar{p}} + W_{\bar{p}}x)]$, where "\|\|" denotes concatenation and $\sigma$ denotes the standard activation function, such as ReLU and GELU. Since $\sin$ and $\cos$ functions also satisfy the required conditions of being non-constant, continuous, and non-linear activation functions, the FAN layer adheres to the Universal Approximation Theorem.
> >
> > We never argue that FAN mitigates spectral bias in our paper and are curious about the origins of this problem. Spectral bias refers to the tendency of neural networks to learn low-frequency (smooth) functions more readily than high-frequency (rapidly changing) ones. Here, our answer is that by incorporating sinusoidal activations like $\sin$ and $\cos$, the FAN layer can represent high-frequency components more effectively, which helps mitigate spectral bias. The inclusion of sinusoidal functions allows the network to capture a broader range of frequencies, enhancing its ability to approximate functions with high-frequency content.

---

> > ### Author Response · Authors · 2024-11-19
> > **Response to Reviewer 7Gdo (Part 3/3)**
> >
> > > Key experimental details such as hyperparameter tuning, model initialization, and evaluation metrics are missing or inadequately described.
> >
> > We clearly stated, "More experimental details and comprehensive setups of each task can be found in Appendix C," and provided the reminder in the main text of our original paper.
> >
> > > There is insufficient analysis of why FAN performs better, and the results for non-periodic tasks suggest that FAN’s benefits might be overstated.
> >
> > We respectfully disagree. We provide the analysis of why FAN performs better in the Discussion Section of our original paper. First, FAN explicitly encodes periodic patterns within its network, naturally endowing it with capabilities for periodicity modeling. Therefore, for periodic tasks and some non-periodic tasks that is partially periodic, FAN leverages its effective periodicity modeling ability to yield better results. Second, many machine learning tasks may harbor hidden forms of periodicity, even without explicit requirements to include periodicity. For instance, consider addition and multiplication operations: despite being strictly non-periodic functions, they involve bit-level operations and carry mechanisms that exhibit periodic characteristics. Therefore, FAN’s ability to capture the latent periodic features may contributes to its enhanced performance in these tasks. The conclusion stems from our experiments, where using FAN exhibits clear improvement in extensive experiments on Symbolic Formula Representation tasks, Time Series Forecasting tasks, and Language Modeling tasks.
> >
> > We think your concerns regarding potential overstatements may stem from your doubt about results in symbolic formula representation. However, as answered in response to Question 2 above, "The experimental results in symbolic formula representation are correct and reproducible, as we directly inherited the code from KAN's project, and our results are consistent with the observations made in KAN as well as some related studies". Therefore, we believe that the conclusion is not overstated, and we will double-confirm our statement to ensure clarity and precision.

---

> > ### Author Response · Authors · 2024-11-21
> > **Looking forward to your feedback**
> >
> > Hello, do our responses address your questions? Please let us know if there are any other questions you'd like to discuss!

---

> > > ### Comment · Reviewer_7Gdo · 2024-11-26
> > >
> > > Hi I have updated the score.
> > > While the authors have addressed some technical concerns, the paper's fundamental contribution requires strengthening before publication.

---

> ### Author Response · Authors · 2024-11-28
>
> Dear Reviewer 7Gdo,
>
> Thank you for your feedback. **We are pleased to hear that your technical concerns have been addressed.** However, regarding your comment on the fundamental contribution of the paper, we find it difficult to understand **what specific areas of our work require to further strengthen**, as the feedback seems somewhat general. We would greatly appreciate it if you could kindly provide more detailed guidance on the aspects that you feel need improvement, so that we can make the necessary revisions and clarify any remaining concerns.
>
> We look forward to your feedback and will make every effort to ensure the paper meets the expectations for publication.
>
> Sincerely,
> Authors

---

### Author Response · Authors · 2024-12-02
**Main Contributions and Revisions**

Thanks for the time and effort invested by all reviewers and chairs. We would like to clarify our main contributions and summarize the revisions made in response to your feedback as follows.

**First, the main contributions of our work can be summarized as follows:**
* We reveal that existing general neural networks, e.g., MLP and Transformer, exhibit potential flaws in the modeling and reasoning of periodicity, especially in OOD scenarios.
* We propose a novel network architecture, termed FAN, based on Fourier analysis, empowering the ability to effectively capture and model periodicity while maintaining general-purpose ability.
* FAN is defined following two core principles: 1) its periodicity modeling capability scales with network depth and 2) the periodicity modeling available throughout the network, thus serving as a building block for deep neural networks.
* FAN can seamlessly replace MLP in various model architectures with fewer parameters and FLOPs, being a promising substitute to MLP.
* Extensive Experiments show that FAN achieves superior results in periodicity modeling tasks and showcases effectiveness and generalizability across a variety of real-world tasks, including symbolic formula representation, time series forecasting, language modeling, and image recognition. Using FAN achieves clear improvements in these tasks, especially in OOD scenarios.

**Second, we have uploaded a revision of the paper by incorporating further clarifications in response to the reviewers' feedback, which mainly includes the following aspects:**
* We have revised the Abstract and Introduction, emphasizing that the design of FAN is not a simple modification of the activation function. Instead, on the basis of introducing Fourier analysis, FAN is founded on two additional core principles, making it have a powerful periodicity modeling ability while maintaining the general-purpose ability.
* We have added additional experiments on Image Recognition in Section 4.5 to further demonstrate the generalizability and effectiveness of FAN, and using FAN achieves superior performance in OOD scenarios.
* We have added the Further Analysis of FAN in Section 4.6, including the impact of hyperparameter $d_p$ and the runtime of FAN, providing deeper insights into its efficiency.
* We have added the additional experiments in Appendix B.2-B.7, including FAN for Solving SciML Problems, further experiments on time series problems, the effectiveness of FAN Layer for deep neural networks, and so on.

---

### Author Response · Authors · 2024-12-02
**Looking forward to the feedback of reviewers**

As the author-reviewer discussion period draws to a close, we kindly encourage the reviewers to acknowledge our rebuttal and pose any follow-up questions they might have. Your thoughtful engagement is highly appreciated.

---

> ### Author Response · Authors · 2024-12-04
>
> We are pleased that the reviewers' technical concerns have been addressed and their recognition of FAN's value during the rebuttal period. We appreciate that if the reviewers could offer any specific suggestions for further enhancing our work or consider updating the score if all questions have been resolved. Thanks for the time and effort of the reviewers and chairs!

---

### Meta-Review · Area_Chair_UEkd · 2024-12-19

**Metareview:**

The paper presents a Fourier Analysis Network (FAN) to mitigate the inherent limitation of MLP and transformers of being unable to model periodicity in data. The method is novel and is intuitive considering periodic nature of fourier analysis.

The paper suffers from the following weaknesses:
1. Practical usability of the method need to be emphasized through evaluations on relevant data sets [Reviewer 7Gdo]
2. Authors make some claims that requires strong justification such as the following:
       - FANs mitigating spectral bias - they refute their claim and then try to make sweeping justifications;
       - missing comparison with KAN approaches
       - Evaluation on sufficient non-periodic data sets
      - Inconsistent baselines across different data sets
      - theoretical justification for why FAN enhances generalization capability.

**Additional Comments On Reviewer Discussion:**

Almost all reviewers agree that the paper will benefit from more theoretical and empirical evaluations, re-organization of the paper:

Reviewer RaTJ :  the existing experiments do not sufficiently demonstrate the novelty or performance required for me to assign a higher score.
Reviewer D8kJ:  I appreciate the authors' reply, and I can see how this approach can have value, but the current experiments are not sufficient to convince me that this approach has the novelty or performance needed for me to assign a higher score.
I would suggest that, in future work, the authors keep the existing motivation (NNs can't learn periodic functions), and then segue into a more specific problem where this periodic limitation leads to clear failure of existing approaches. The Burgers' equation results with FNOs the authors mention in the rebuttal period would be a great motivating example, as would a more extensive time series problem with a periodic component, like physiological data or EEGs (current forecast models still underperform in this domain, perhaps due to the periodicity issue identified by the authors). The language modeling example still doesn't seem well-justified to me, and would be better suited for an appendix.
Reviewer 7Gdo: While the authors have addressed some technical concerns, the paper's fundamental contribution requires strengthening before publication.

---

### Decision · Program_Chairs · 2025-01-22

Reject